# VASEVQA-3D: BENCHMARKING 3D VLMS ON ANCIENT GREEK POTTERY

**Nonghai Zhang**[1*] **Zeyu Zhang**[1*†] **Jiazi Wang**[2*] **Yang Zhao**[3] **Hao Tang**[1‡]

[1]School of Computer Science, Peking University
[2]School of Computer Science and Technology, Beijing Jiaotong University
[3]La Trobe University

[*]Equal contribution. [†]Project lead. [‡]Corresponding author: bjdxtanghao@gmail.com

## ABSTRACT

Vision-Language Models (VLMs) have achieved significant progress in multimodal understanding tasks, demonstrating strong capabilities particularly in general tasks such as image captioning and visual reasoning. However, when dealing with specialized cultural heritage domains like 3D vase artifacts, existing models face severe data scarcity issues and insufficient domain knowledge limitations. Due to the lack of targeted training data, current VLMs struggle to effectively handle such culturally significant specialized tasks. To address these challenges, we propose the **VaseVQA-3D** dataset, which serves as the first 3D visual question answering dataset for ancient Greek pottery analysis, collecting 664 ancient Greek vase 3D models with corresponding question-answer data and establishing a complete data construction pipeline. We further develop the **VaseVLM** model, enhancing model performance in vase artifact analysis through domain-adaptive training. Experimental results validate the effectiveness of our approach, where our VaseVLM-7B-RL achieves 12.8% improvement in R@1 accuracy and 6.6% improvement in lexical similarity compared to the strongest baselines on the VaseVQA-3D dataset, significantly improving the recognition and understanding of 3D vase artifacts, providing new technical pathways for digital heritage preservation research. Code: `https://github.com/AIGeeksGroup/VaseVQA-3D`. Website: `https://aigeeksgroup.github.io/VaseVQA-3D`.

## 1 INTRODUCTION

A picture is worth a thousand words. Vision-Language Models (VLMs) have demonstrated remarkable capabilities in understanding and reasoning about visual content, achieving impressive performance across various multimodal tasks such as Visual Question Answering (VQA), image captioning, and visual reasoning tasks (Liu et al., 2023a; Deitke et al., 2025). These models have shown significant progress in bridging the gap between visual perception and language understanding, performing excellently in general-scenario visual question answering tasks and providing new possibilities for handling complex visual understanding tasks. However, existing VLMs face two

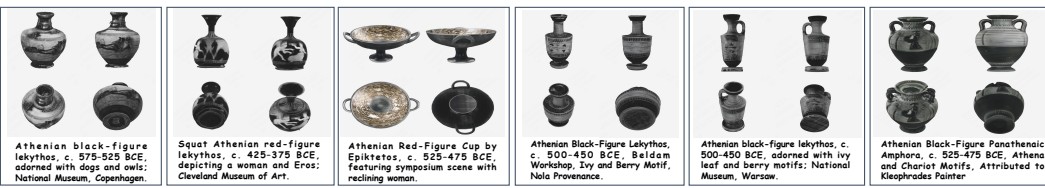

Figure 1: **Captions in VaseVQA-3D dataset**. Each GLB-format 3D vase is rendered in four canonical views—front, back, top, and bottom—and is accompanied by a concise caption that records decorative motifs, manufacturing technique, provenance, and current repository

significant challenges when dealing with specialized domains like 3D vase artifacts: (1)Ancient Greek pottery, as a crucial carrier of Hellenic civilization with over 2,000 years of historical significance and more than 300 years of systematic archaeological research, represents rare long-tail data for VLMs, yet existing datasets lack comprehensive 3D representations of these artifacts, and (2) due to data scarcity, no current VLMs can handle such specialized tasks competently, yet this domain is crucial for digital heritage preservation and cultural heritage transmission, creating an urgent need for specialized vase-domain VLMs.

To address the challenge of 3D vase artifact data scarcity, we need to construct specialized 3D vase datasets and benchmarks that can systematically assess model performance on 3D cultural heritage objects. Specifically, ancient Greek pottery requires 3D understanding because its archaeological significance lies in spatial features (symmetry, proportions, morphology) and complete geometric representation, which cannot be fully captured through fragmented 2D views. To address the challenge of insufficient model expertise, we need to develop VLMs with vase-domain expertise, enhancing model capabilities in archaeological analysis tasks through domain-adaptive training.

We introduce **VaseVQA-3D**, a specialized dataset for ancient Greek pottery visual question answering, addressing the data gap in this professional domain. Our question-answer pairs are structured around six archaeological semantic dimensions (Fabric, Technique, Shape, Dating, Decoration, Attribution), reflecting professional archaeological knowledge rather than generic visual attributes. We propose a comprehensive pipeline for transforming existing 2D vase images into high-fidelity 3D representations, including rigorous data filtering, 2D-to-3D conversion techniques, and large model enhancement of existing vase QA data. We collect 24 high-quality real GLB models as *VaseEval* to systematically evaluate the effectiveness of our data synthesis pipeline. Based on the validated high-quality data, we develop **VaseVLM**, a VLM specifically fine-tuned for 3D vase understanding capabilities through domain-adaptive training strategies and archaeological expertise integration. We conduct extensive evaluations on multiple state-of-the-art VLMs, and experimental results demonstrate that our fine-tuned model can achieve better recognition and understanding of 3D vase data, validating the effectiveness of our approach in 3D vase and other cultural heritage domains, providing new technical perspectives and solutions for digital heritage preservation.

In summary, the contributions of our paper can be summarized in three folds:

- We introduce **VaseVQA-3D**, a comprehensive dataset for evaluating VLMs on 3D ancient pottery, including 664 high-quality 3D vase models and diverse question-answer pairs exploring vase attribute information. We also construct VaseEval for evaluating 3D asset quality, filling the data gap in this professional domain.
- We propose **VaseVLM**, a vision-language model specifically fine-tuned for 3D vase understanding. Since 3D vase artifacts are rare and constitute long-tail data, existing VLMs struggle with such specialized tasks. Our VaseVLM employs a two-stage training approach: first establishing baseline performance through LoRA-based supervised fine-tuning (SFT) on 360-degree rotation videos and archaeological captions, then applying GRPO reinforcement learning with our novel Reinforcement Learning with Verifiable Rewards (RLVR) framework that decomposes archaeological descriptions into six semantic dimensions (Fabric, Technique, Shape, Dating, Decoration, Attribution) for multi-dimensional reward computation and quality control.
- We conduct comprehensive experimental evaluation demonstrating significant improvements in archaeological VQA tasks. Our VaseVLM-7B-RL achieves 12.8% improvement in R@1 accuracy and 6.6% improvement in lexical similarity compared to the strongest baselines on the VaseVQA-3D dataset. Beyond technical contributions, our work has important social value in digital heritage preservation, providing meaningful exploration for AI technology applications in cultural heritage protection and offering new pathways for digital heritage analysis and interdisciplinary collaboration between computer science and archaeology.

## 2 RELATED WORK

**VLMs and Visual Question Answering.** VLMs serve as core technology in multimodal AI, enabling machines to understand and reason about visual content through natural language. Modern VLM development is grounded in contrastive learning, with CLIP (Radford et al., 2021) having pioneered large-scale visual-text alignment. Recent large VLMs have significantly enhanced visual understanding capabilities. Closed-source models like GPT-4V (Hurst et al., 2024) and Gem-

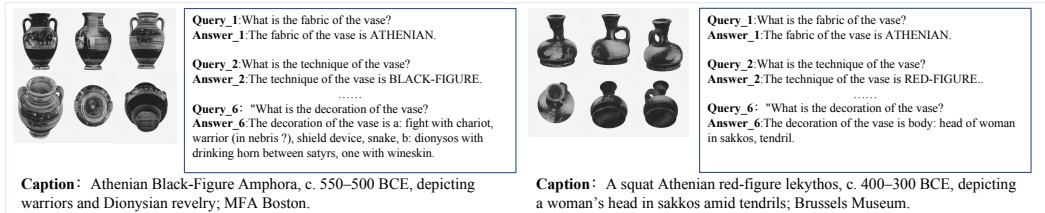

Figure 2: **QA in VaseVQA-3D dataset.** Each data entry contains high-quality 3D vase models, structured question-answer pairs, and GPT-4o enhanced descriptive captions, providing comprehensive support for multimodal understanding of ancient Greek pottery.

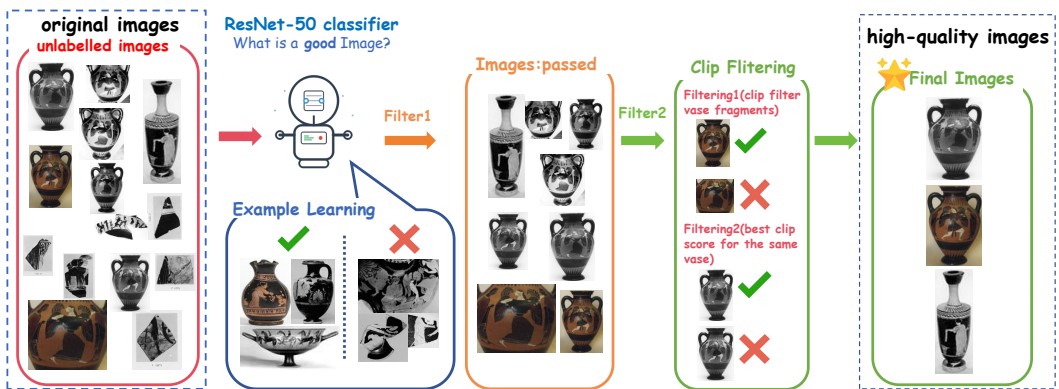

Figure 3: **Complete Data Quality Filtering Pipeline.** The figure shows our comprehensive filtering methodology, including ResNet-50-based quality assessment for removing low-quality images, followed by dual CLIP-based semantic filtering for fragment removal and optimal image selection.

ini (Comanici et al., 2025), and open-source models such as Qwen2.5-VL (Bai et al., 2025) and InternVL (Chen et al., 2024) have demonstrated remarkable performance across various multimodal tasks. Specialized techniques have emerged in 3D vision-language understanding: Cap3D (Luo et al., 2023), DiffuRank (Luo et al., 2024), and LLaVA-3D (Zhu et al., 2024) achieved advances in 3D model descriptions and question-answering.

**3D Generation and Reconstruction.** In image-to-3D model generation, DreamFusion (Poole et al., 2022) pioneered the application of image diffusion priors to 3D generation. Recent advances include TripoSG (Li et al., 2025) and Hunyuan3D (Lai et al., 2025), which achieved state-of-the-art performance in 3D shape generation tasks through improved architectures and training strategies.

**Cultural Heritage and Archaeological AI.** AI technologies are effectively enhancing cultural heritage preservation. Recent works include ArchaeoScape (Perron et al., 2024) for archaeological site identification and automated restoration methods for cultural artifacts (Feng et al., 2025).

For detailed related work introduction, please refer to Appendix A.2.

## 3   DATASET: VASEVQA-3D

**Data Sources and Collection**   Our VaseVQA-3D dataset construction is based on two main data sources: the large-scale 2D vase image collection and corresponding archaeological metadata provided by the VaseVQA (Ge et al., 2025) dataset (as shown in Figure 5), and a curated set of high-quality 3D references drawn from the Sketchfab digital museum (as shown in Figure 4). The VaseVQA dataset serves as our primary data source, containing over 30,000 2D images of ancient Greek vases, each accompanied by detailed archaeological metadata annotations covering six core vase attributes: fabric composition, manufacturing techniques, morphological shapes, historical dating, decorative elements, and artistic attribution. To validate our 3D generation pipeline quality and perform generative model selection, we collected high-quality reference data from the Sketchfab digital museum to construct our VaseEval validation set for 3D generation quality assessment.

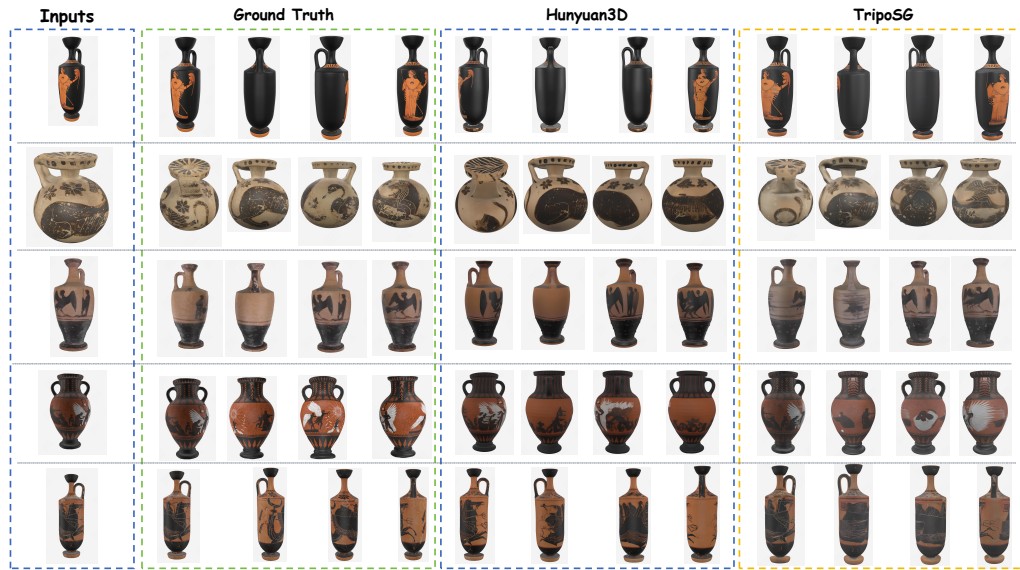

Figure 4: 3D Generation Methods Comparison. Comparison of TripoSG and Hunyuan3D generation effects based on the VaseEval validation set. TripoSG performs better in mesh quality, and although Hunyuan3D has advantages in texture mapping effects, TripoSG-generated models are closer to ground truth, thus selected for large-scale dataset construction.

**Data Quality Filtering** However, the original VaseVQA dataset suffers from significant quality issues, containing numerous vase fragments, blurred images, and even sketches, which severely impact the dataset's suitability for high-quality 3D generation tasks. To enhance dataset quality, we designed a three-stage progressive filtering framework, as shown in Figure 3. First, we train a ResNet-50 binary classifier based on manual annotations for preliminary quality screening, automatically identifying and removing low-quality images. Second, we employ CLIP models for fragment detection, using predefined text prompts to calculate the similarity difference between images and descriptions of complete vases versus fragments, adopting a binary classification approach to automatically identify and remove vase fragments. Finally, addressing the multi-viewpoint issue for each vase, we use CLIP to compute similarity scores between each viewpoint image and high-quality descriptive text, selecting the image with the highest score as the optimal representative view. This comprehensive filtering mechanism ensures the final dataset contains only high-quality, complete, and archaeologically representative vase images, providing a reliable foundation for subsequent 3D generation and VQA tasks.

**VaseEval: 3D Generation Quality Assessment** To ensure the quality and reliability of our 3D generation pipeline, we construct VaseEval(as shown in Figure 4), a specialized validation set consisting of 24 high-quality ancient Greek vase GLB files carefully selected from the Sketchfab digital museum. These reference models serve as ground truth for evaluating the effectiveness of different 3D generation methods and validating our data synthesis pipeline.

As shown in Figure 4. VaseEval covers diverse vase morphologies including narrow-body vases, wide-mouth vessels, and various decorative patterns, providing comprehensive coverage for quality assessment. Each model in VaseEval features clear geometric structures and rich textural details, enabling systematic evaluation of both mesh quality and texture fidelity in generated 3D models.

Through comparative analysis using VaseEval, we validated our choice of 3D generation method and ensured the archaeological accuracy and visual quality of our final VaseVQA-3D dataset. VaseEval served as the benchmark for selecting the most suitable 3D generation method, ensuring that our chosen approach produces high-quality 3D models with accurate geometric representation and visual fidelity for ancient Greek pottery.

**VaseVQA-3D Construction** Based on the filtered high-quality 2D images, we designed a comprehensive data synthesis pipeline that converts 2D images into high-fidelity 3D models while main-

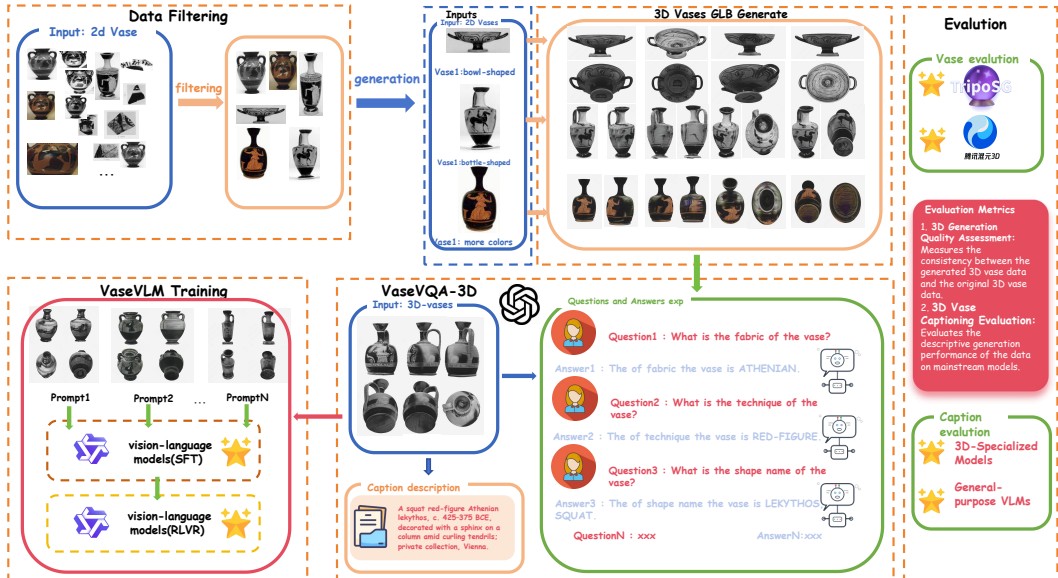

Figure 5: Complete Pipeline for Vase Dataset Construction. The pipeline progresses from initial data collection (30K+ images) through quality filtering (3,880 images), 3D generation (664 models), QA construction (4K+ pairs), to final model training. Each component includes specific quality control mechanisms and validation procedures.

taining archaeological accuracy and cultural authenticity, as shown in Figure 5. Our data synthesis pipeline contains three core stages: first, filtering 3,880 high-quality 2D images from 30,000+ original images through ResNet-50 and CLIP dual filtering mechanisms; then, converting these 2D images into 664 high-fidelity 3D models using TripoSG technology; and finally, generating 4,460 structured question-answer pairs and corresponding descriptive captions by cleaning and organizing the original archaeological metadata using GPT-4o (Hurst et al., 2024).

To construct high-quality 3D vase models, we evaluated two currently recognized superior 2D-to-3D generation methods: TripoSG and Hunyuan3D, both of which excel in 2D-to-3D reconstruction tasks. We used the VaseEval validation set to evaluate the effectiveness of both generation methods by capturing front-view photographs of these 3D models, as shown in Figure 4. From the analysis of the results in the figure, we found that TripoSG generated higher mesh quality, while Hunyuan3D produced better texture mapping effects. However, to restore the ground truth effect, we generally believed that TripoSG generated more realistic results with better vase model quality, so we ultimately chose TripoSG for large-scale data generation.

As shown in Figure 2. Our VaseVQA-3D dataset comprises two complementary components. The structured VQA component directly adopts the original question-answer content from VaseVQA, covering six core vase attributes: Fabric, Technique, Shape, Dating, Decoration, and Attribution. Each question follows the standardized format "What is the [attribute] of the vase?" to ensure consistency and fairness in evaluation. The answers are derived from verified archaeological metadata, maintaining factual accuracy and scholarly reliability.

The caption component provides descriptive captions for each vase by organizing and cleaning the original archaeological metadata from the VaseVQA dataset. The original metadata contains structured information (e.g., "Fabric: ATHENIAN; Technique: BLACK-FIGURE; Shape: LEKYTHOS; Date: -525 to -475 BCE") but is often fragmented and noisy. We use GPT-4o to consolidate this existing archaeological information into coherent museum-style descriptions (e.g., "Athenian black-figure lekythos, c. 525–475 BCE") without introducing new archaeological content. This cleaning process removes noise and improves readability while preserving the original archaeological facts. The final VaseVQA-3D dataset contains GLB-format 3D model files, with each model associated with corresponding structured question-answer pairs and cleaned descriptive captions, providing a solid foundation for training and evaluating VLMs in the cultural heritage domain.

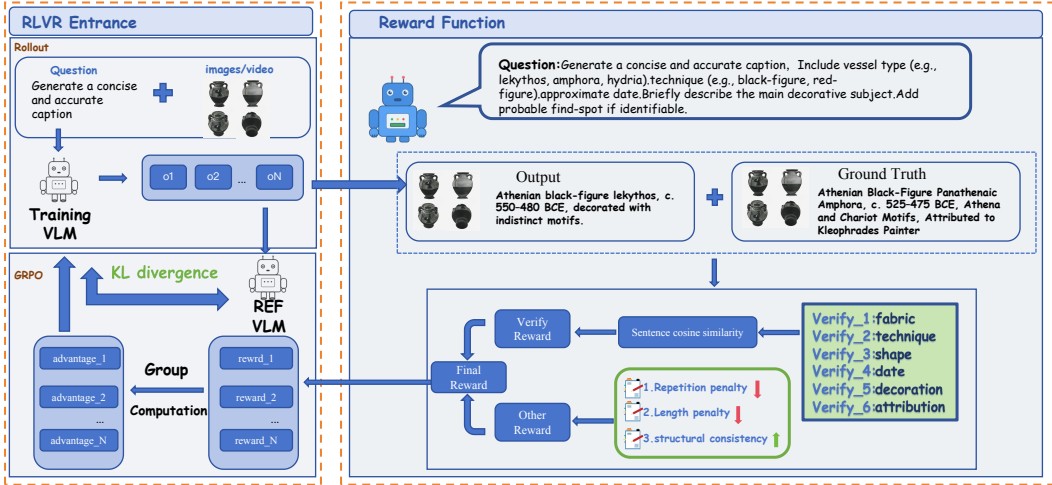

Figure 6: Reinforcement Learning with Verifiable Rewards (RLVR) Framework. The figure shows our multi-dimensional reward computation system that evaluates archaeological descriptions across six semantic dimensions: Fabric, Technique, Shape, Dating, Decoration, and Attribution. The framework includes semantic similarity analysis, quality control penalties, and similarity rewards to ensure accurate and academically appropriate responses.

## 4 THE PROPOSED METHOD: VASEVLM

**Overview** This section introduces the comprehensive pipeline of our 3D visual question answering system specifically designed for ancient Greek vase analysis, as shown in Figure 5. Our method contains several key components: initial data collection from large-scale 2D vase datasets, comprehensive data quality filtering combining deep learning classifiers and CLIP-based methods, 3D generation method validation and large-scale model generation, refining and organizing content based on original VaseVQA QA data using GPT-4o to obtain concise and clear caption question-answer data to improve QA quality and construct the final VaseVQA-3D, dataset quality evaluation using multiple VLM baselines, and specialized model training using fine-tuning and reinforcement learning.

**Supervised Fine-Tuning (SFT)** Based on the constructed VaseVQA-3D dataset, we further trained a specialized ancient Greek vase analysis model VaseVLM and evaluated the dataset's performance across different VLMs. We adopt Qwen2.5-VL (3B and 7B variants) as the base model, with training inputs comprising 360-degree rotation videos generated from GLB files and corresponding refined caption descriptions that contain rich archaeological information with concise expression. We establish the VaseVLM baseline model through LoRA-based (Hu et al., 2022) supervised fine-tuning (SFT) (Ouyang et al., 2022).

**Reinforcement Learning (RL)** We later employ the GRPO (Shao et al., 2024) reinforcement learning method for further optimization. Our VaseVQA-3D dataset is naturally suited for RLVR (Lambert et al., 2024) training, as the ground truth captions contain complete information across six dimensions (Fabric, Technique, Shape, Dating, Decoration, and Attribution). During the GRPO rollout phase, the model generates captions containing this dimensional information, which are then compared and verified against the standard answers in the ground truth, analogous to mathematical problem-solving processes where the model generates answers for comparison with standard solutions.

$$r_i = \begin{cases} \mathrm{sim}(g_i, t_i), & \text{if } \mathrm{sim}(g_i, t_i) \geq \tau \\ 0, & \text{otherwise} \end{cases} \tag{1}$$

As shown in Figure 6, our GRPO strategy employs Reinforcement Learning with Verifiable Rewards (RLVR) to compute the reward function. This approach decomposes archaeological descriptions into six core semantic dimensions for evaluation, each with corresponding weights: Fabric ($w_f = 0.20$),

Table 1: CLIP-based Data Quality Filtering Results.

| Filtering Stage | Input Images | Output Images | Retention Rate | Quality Score |
|---|---|---|---|---|
| Initial Collection | 30,000 | 30,000 | 100% | - |
| Resnet-50 Quality Filtering | 30,000 | 13,599 | 45.3% | - |
| CLIP Fragment Filtering | 13,599 | 6,330 | 46.5% | 0.156 |
| CLIP View Selection | 6,330 | 3,880 | 61.3% | 0.234 |
| 3D Generation (TripoSG) | 3,880 | 664 | 17.1% | - |
| **Overall Pipeline** | **30,000** | **664** | **2.2%** | **0.234** |

Table 2: 3D Generation Methods Comparison on 24 Ground Truth Models.

| Method | PSNR↑ | SSIM↑ | LPIPS↓ | CD↓ | NC↑ | CLIP-I↑ | CLIP-T↑ |
|---|---|---|---|---|---|---|---|
| **Reference** | **15-25** | **0.7-0.9** | **0.1-0.3** | **0.1-0.3** | **0.6-0.8** | **0.7-0.9** | **0.6-0.8** |
| TripoSG (Li et al., 2025) | 17.21 | **0.8676** | **0.1308** | **0.1490** | 0.7232 | **0.8896** | **0.9594** |
| Hunyuan3D (Zhao et al., 2025) | **17.23** | 0.8657 | 0.1319 | 0.1515 | **0.7389** | 0.8837 | 0.9237 |

Technique ($w_t = 0.20$), Shape ($w_s = 0.15$), Dating ($w_d = 0.15$), Decoration ($w_{dec} = 0.20$), and Attribution ($w_a = 0.10$). The dimensional reward $r_i$ is calculated as:

where $g_i$ and $t_i$ represent the generated content and target content for dimension $i$, $\text{sim}(\cdot, \cdot)$ denotes cosine similarity, and $\tau = 0.7$ is the similarity threshold. In addition to content accuracy assessment, our reward function includes quality control penalty mechanisms:

$$P = \alpha_l P_{\text{length}} + \alpha_r P_{\text{repetition}} + \alpha_i P_{\text{irrelevant}}, \tag{2}$$

which penalizes inappropriate length, repetitive phrasing, and irrelevant content respectively, with penalty weights set to $\alpha_l = 0.1$, $\alpha_r = 0.1$, and $\alpha_i = 0.15$. The final reward function is:

$$R = \sum_{i=1}^{6} w_i \cdot r_i - P + B, \tag{3}$$

where $B$ is a sequence matching-based similarity reward term, and the complete reward is constrained to the unit interval $[0, 1]$ to ensure training stability during policy optimization.

Through this complete pipeline of data construction, model training, and evaluation, we successfully established an end-to-end system from 2D images to 3D models to specialized VQA models, providing an effective technical solution for intelligent analysis of ancient Greek pottery.

## 5 EXPERIMENTS

This section presents a comprehensive experimental evaluation of our pipeline for constructing VaseVQA-3D datasets and training specialized models for ancient Greek vase analysis. We conduct extensive experiments to validate the effectiveness of each component in our methodology, compare against state-of-the-art baselines, and provide a detailed analysis of the results across multiple evaluation dimensions. Our experiments focus on three key aspects: CLIP-based data quality filtering, 3D generation method comparison, and dataset quality assessment.

### 5.1 DATASETS AND EVALUATION METRICS

**Experimental Datasets.** This study uses three datasets for experimental validation at different stages. VaseVQA Original Dataset contains 30,000 ancient Greek vase images as the input data source for our pipeline. VaseVQA-3D is the core dataset after quality filtering and 3D generation, containing 664 3D vase models and 4,460 question-answer pairs, divided into training set (464 models), validation set (100 models), and test set (100 models) with a 70%/15%/15% split. VaseEval contains 24 professional GLB files collected from Sketchfab as ground truth references for 3D generation method evaluation.

**Evaluation Metrics.** We adopt a three-category evaluation metric system. *3D Generation Quality Metrics* are used for VaseEval validation, including visual quality metrics (PSNR (Wang et al.,

Table 3: Comprehensive Dataset Quality Assessment Results by Individual Models.

| Method | FID↓ | CLIP↑ | R@10↑ | R@5↑ | R@1↑ | Lexical Sim.↑ |
|---|---|---|---|---|---|---|
| *3D-Specialized Models* | | | | | | |
| DiffuRank (Luo et al., 2024) | 0.421 | **0.798** | 16.67% | 8.33% | 2.08% | 0.274 |
| Cap3D (Luo et al., 2023) | 0.445 | 0.792 | 14.58% | 7.29% | 1.56% | 0.267 |
| LLaVA3D (Zhu et al., 2024) | 0.494 | 0.784 | 10.42% | 5.21% | 1.04% | 0.238 |
| *Closed-source VLMs* | | | | | | |
| Gemini-2.5-flash (Comanici et al., 2025) | **0.325** | 0.736 | **28.57%** | **17.58%** | 2.20% | 0.210 |
| Claude-4-sonnet (Anthropic, 2025b) | 0.353 | 0.676 | 23.96% | 10.42% | 3.12% | 0.188 |
| Gemini-2.5-Pro (Comanici et al., 2025) | 0.397 | 0.680 | 22.92% | 14.58% | 3.12% | 0.162 |
| GPT-4.1 (OpenAI, 2025) | 0.501 | 0.644 | 25.00% | 10.42% | 3.12% | 0.128 |
| Claude-3.5-sonnet (Anthropic, 2024) | 0.455 | 0.643 | 15.62% | 8.33% | 2.08% | 0.116 |
| Doubao-1.5-vision-pro-32k (ByteDance, 2025) | 0.504 | 0.606 | 14.58% | 4.17% | 1.04% | 0.074 |
| GPT-4o (Hurst et al., 2024) | 0.582 | 0.520 | 13.54% | 6.25% | 2.08% | 0.104 |
| Claude-3.7-sonnet (Anthropic, 2025a) | 0.600 | 0.339 | 13.54% | 6.25% | 1.04% | 0.101 |
| *Open-source VLMs* | | | | | | |
| InternVL (Chen et al., 2024) | 0.376 | 0.771 | 10.42% | 8.33% | 2.08% | 0.252 |
| Qwen2.5-VL-7B (Bai et al., 2025) | 0.334 | 0.775 | 18.75% | 9.38% | 2.08% | 0.217 |
| Qwen2.5-VL-3B (Bai et al., 2025) | 0.358 | 0.782 | 9.38% | 6.25% | 1.04% | 0.259 |
| VaseVL (Ge et al., 2025) | 0.493 | 0.790 | 10.4% | 6.25% | 2.08% | 0.255 |
| **VaseVLM-3B-SFT (Ours)** | 0.359 | 0.788 | 17.71% | 8.33% | 2.08% | 0.223 |
| **VaseVLM-3B-RL (Ours)** | 0.363 | 0.789 | 17.71% | 10.42% | 2.08% | 0.245 |
| **VaseVLM-7B-SFT (Ours)** | 0.332 | 0.779 | 20.83% | 10.42% | 3.12% | 0.272 |
| **VaseVLM-7B-RL (Ours)** | 0.328 | 0.792 | 21.24% | 11.12% | **3.52%** | **0.276** |

2004), SSIM (Hore & Ziou, 2010), LPIPS (Johnson et al., 2016)), geometric accuracy metrics (Chamfer Distance(CD) (Fan et al., 2017b), Normal Consistency(NC) (Li et al., 2023a)), and semantic consistency metrics (CLIP Image/Text Similarity) (Radford et al., 2021). *VQA Capability Evaluation Metrics* are used for model performance assessment, including FID Score (Heusel et al., 2017), CLIP Score (Shen et al., 2021), retrieval metrics R@1/5/10 (Fang et al., 2015), and lexical similarity (Lin, 2004). *Human Evaluation Metrics* employ 10 experts to score model-generated captions on a 0-5 scale, evaluating description accuracy and cultural appropriateness.

## 5.2 IMPLEMENTATION DETAILS

**Training Setup.** All experiments are conducted on a high-performance computing cluster equipped with NVIDIA A100 GPUs. The hardware configuration includes $8\times$ NVIDIA A100 GPUs (80GB VRAM each), $2\times$ Intel Xeon Platinum 8358 CPUs (32 cores each), 1TB DDR4 memory, and 10TB NVMe SSD storage. The total computational time for the entire experimental pipeline is approximately 14.5 days on a single A100, including 13.5 days for 3D generation, 4 hours for supervised fine-tuning, and 20 hours for reinforcement learning training. For detailed hyperparameters and implementation settings, please refer to Section A.4.

**Experimental Workflow.** We adopt a three-stage progressive mechanism to construct our VaseVQA-3D dataset and train specialized models. The first stage involves data filtering using ResNet-50 and CLIP-based quality assessment to select high-quality vase images from the original VaseVQA dataset. The second stage focuses on 3D generation and dataset construction, where we employ TripoSG for single-image 3D reconstruction and generate corresponding video sequences with enhanced captions. The third stage encompasses model training and evaluation, including supervised fine-tuning and reinforcement learning optimization of VaseVLM variants. For detailed technical implementation, please refer to Appendix A.5.

## 5.3 EXPERIMENTAL RESULTS

Our comprehensive experimental evaluation demonstrates the effectiveness of each component in our pipeline for constructing high-quality VaseVQA-3D datasets and training specialized models for ancient Greek vase analysis. The results validate our approach across multiple dimensions, including data quality filtering, 3D generation method selection, and dataset quality assessment.

**Data Quality Filtering Effectiveness Analysis.** Based on the results in Table 1, our three-stage progressive filtering strategy demonstrates high selectivity and effectiveness. Starting from 30,000

Table 4: Human Evaluation Results: Expert Ratings for Vase Caption Generation (Scale: 0-5).

| Method | Exp-1 | Exp-2 | Exp-3 | Exp-4 | Exp-5 | Exp-6 | Exp-7 | Exp-8 | Exp-9 | Exp-10 | Ave. | Rank |
|---|---|---|---|---|---|---|---|---|---|---|---|---|
| *Fine-tuned Models (Ours)* | | | | | | | | | | | | |
| VaseVLM-7B-RL | 4.6 | 4.8 | 4.5 | 4.7 | 4.4 | 4.6 | 4.5 | 4.4 | 4.7 | 4.5 | 4.57 | 1 |
| VaseVLM-3B-RL | 4.4 | 4.6 | 4.3 | 4.5 | 4.2 | 4.4 | 4.3 | 4.2 | 4.5 | 4.3 | 4.37 | 2 |
| VaseVLM-7B-SFT | 4.2 | 4.4 | 4.1 | 4.3 | 4.0 | 4.2 | 4.1 | 4.0 | 4.3 | 4.1 | 4.17 | 3 |
| VaseVLM-3B-SFT | 4.0 | 4.2 | 3.9 | 4.1 | 3.8 | 4.0 | 3.9 | 3.8 | 4.1 | 3.9 | 3.97 | 4 |
| *3D-Specialized Models* | | | | | | | | | | | | |
| DiffuRank | 4.1 | 4.3 | 4.0 | 4.2 | 3.9 | 4.1 | 4.0 | 3.9 | 4.2 | 4.0 | 4.07 | 5 |
| *General-purpose VLMs* | | | | | | | | | | | | |
| Gemini-2.5-flash | 3.9 | 4.1 | 3.8 | 4.0 | 3.7 | 3.9 | 3.8 | 3.7 | 4.0 | 3.8 | 3.87 | 6 |
| VaseVL | 3.8 | 4.0 | 3.7 | 3.9 | 3.6 | 3.8 | 3.7 | 3.6 | 3.9 | 3.7 | 3.77 | 7 |
| Claude-4-sonnet | 3.7 | 3.9 | 3.6 | 3.8 | 3.5 | 3.7 | 3.6 | 3.5 | 3.8 | 3.6 | 3.67 | 8 |
| Qwen2.5-VL-7B | 3.6 | 3.8 | 3.5 | 3.7 | 3.4 | 3.6 | 3.5 | 3.4 | 3.7 | 3.5 | 3.57 | 9 |
| Gemini-2.5-Pro | 3.5 | 3.7 | 3.4 | 3.6 | 3.3 | 3.5 | 3.4 | 3.3 | 3.6 | 3.4 | 3.47 | 10 |
| **Overall Average** | **4.0** | **4.2** | **3.9** | **4.1** | **3.8** | **4.0** | **3.9** | **3.8** | **4.1** | **3.9** | **3.95** | **–** |

initial images, the classifier quality filtering retains 45.3% of images, effectively removing blurry, overly dark, and low-resolution samples. The subsequent CLIP fragment filtering further screens to 46.5%, successfully identifying and removing vase fragments. The CLIP view selection stage has a retention rate of 61.3%, ensuring that only the best viewing angle is retained for each vase, ultimately obtaining 3,880 high-quality images.

The quality score significantly improved from 0.156 after fragment filtering to 0.234 after view selection, an improvement of 50%. After the complete filtering pipeline, the TripoSG 3D generation process successfully produced 664 high-quality GLB models from 3,880 images, with a generation success rate of 17.1%.

**3D Generation Method Comparison Analysis.** To select the optimal 3D generation method for our dataset construction, we conducted a comprehensive comparison between TripoSG and Hunyuan3D using our VaseEval validation set. The evaluation encompasses seven key metrics including image quality, geometric accuracy, and semantic consistency. Table 2 shows that while both methods perform competitively, TripoSG demonstrates superior performance in geometric reconstruction accuracy and semantic understanding, making it more suitable for archaeological VQA applications. For detailed comparative analysis, please refer to Appendix A.6.

**Comprehensive Dataset Quality Assessment.** After confirming the effectiveness of our 3D generation approach, we proceed to generate our complete 3D vase dataset consisting of 664 models. To ensure the quality and reliability of our constructed dataset, as shown in Table 3, we establish a comprehensive evaluation framework using multiple state-of-the-art VLMs as baselines. We evaluate both the QA effectiveness and caption quality of our generated dataset across different model categories, including 3D-specialized models, advanced 3D VLMs, and general-purpose VLMs. For detailed experimental analysis, please refer to Appendix A.7. For detailed examples across VLMs, please refer to Appendix A.9 and A.8.

Our VaseVLM models perform excellently across all metrics, validating the effectiveness of specialized training. VaseVLM-7B-RL achieves the best comprehensive performance (FID: 0.328, CLIP: 0.792, R@10: 21.24%), maintaining low FID while achieving high CLIP scores and retrieval accuracy.

**Performance Comparison and Baseline Selection.** Our VaseVLM-7B-RL demonstrates superior performance across multiple metrics. Specifically, we achieve 3.52% in R@1 accuracy, representing a 12.8% relative improvement over Claude-4-sonnet's 3.12% (the highest R@1 among all baseline models). In lexical similarity, our model achieves 0.276, a 6.6% improvement over Qwen2.5-VL-3B's 0.259 (the best performance in this metric among comparable models). We compare against different baselines for different metrics because no single baseline model excels across all evaluation dimensions, reflecting the multi-faceted nature of archaeological VQA tasks.

It is important to note that 3D-specialized models (DiffuRank, Cap3D, LLaVA3D) operate under fundamentally different task settings: they generate descriptions directly from GLB files with complete 3D geometric information, while our VaseVLM understands 3D content from 2D rotation videos, requiring spatial reasoning and 3D structure inference. These models are included for completeness but are not directly comparable due to different input modalities.

**Human Evaluation Results Analysis.** The expert evaluation results in Table 4 further validate the superiority of our approach. VaseVLM-7B-RL achieved the highest average score of 4.57, receiving consistent recognition from 10 archaeological experts. Reinforcement learning trained models (VaseVLM-7B-RL: 4.57, VaseVLM-3B-RL: 4.37) significantly outperformed supervised fine-tuning versions (VaseVLM-7B-SFT: 4.17, VaseVLM-3B-SFT: 3.97), with an average improvement of approximately 0.4 points, demonstrating the effectiveness of the GRPO method in improving description quality and cultural appropriateness.

Compared to baseline models, our best model surpassed all 3D-specialized models and general-purpose VLMs. Particularly compared to the best-performing baseline DiffuRank (4.07 points), VaseVLM-7B-RL achieved a 12.3% improvement, demonstrating the significant effects of specialized training and reinforcement learning optimization. These results validate our approach of domain-specific training and reinforcement learning optimization. While different baseline models excel in different metrics, our VaseVLM-7B-RL achieves competitive or superior performance across all evaluation dimensions, demonstrating the effectiveness of specialized training for cultural heritage VQA tasks.

**Additional Results and Generalization Analysis.** Beyond the main benchmark and human evaluation reported above, we include supplementary analyses that provide additional context for interpreting the results. These include 3D generation comparisons, model-family analysis across 3D-specialized models, closed-source VLMs, open-source VLMs, and VaseVLM variants, as well as archaeological qualitative cases and representative VLM outputs. Together, these analyses help separate the effects of 3D asset generation, model architecture and scale, and domain-specific training, while also illustrating typical fine-grained archaeological errors such as technique and vessel-type confusion. More importantly, we emphasize that our methodology is not limited to ancient Greek pottery. To validate the generalizability of our pipeline across different cultural heritage domains, we conduct supplementary experiments on Chinese bronze artifacts and ancient Greek sculptures (Appendix A.10). These experiments demonstrate that our end-to-end pipeline can be adapted to other artifact types by customizing domain-specific reward dimensions, providing evidence for the broader applicability of our approach.

## 6 LIMITATIONS AND FUTURE WORK

While our VaseVQA-3D dataset and pipeline demonstrate effectiveness in 3D cultural heritage analysis, several limitations remain. The 17.1% generation success rate from filtered images reflects the difficulty of producing high-quality 3D assets from noisy cultural heritage data. The current pipeline also requires substantial computational resources for 3D generation, video rendering, supervised fine-tuning, and reinforcement learning. Future work should focus on improving 3D generation success rates through stronger reconstruction methods, extending the artifact-oriented pipeline to broader cultural heritage domains, and developing more efficient training strategies that reduce computational cost while preserving domain-specific quality.

## 7 CONCLUSION

This paper introduces VaseVQA-3D, the first 3D visual question answering dataset for ancient Greek pottery analysis, and presents an end-to-end pipeline for cultural heritage VQA. The dataset contains 664 3D vase models and 4,460 question-answer pairs, with VaseEval providing a validation set for 3D generation method selection and quality assessment. Building on this pipeline, which covers data filtering, 3D generation, caption and QA construction, domain-specific post-training, and evaluation, our VaseVLM models demonstrate the effectiveness of specialized training for artifact understanding. VaseVLM-7B-RL achieves 12.8% improvement in R@1 accuracy and 6.6% improvement in lexical similarity over the strongest baselines in their respective metrics. Together with supplementary experiments on other artifact domains, VaseVQA-3D establishes a new benchmark for 3D cultural heritage understanding and offers a practical framework for applying specialized AI systems to digital heritage preservation.

**Acknowledgements.** This work is supported by the Fundamental Research Funds for the Central Universities, Peking University.

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

# A  APPENDIX

## A.1  LLM USE DECLARATION

Large Language Models (ChatGPT) were used exclusively to improve the clarity and fluency of English writing. They were not involved in research ideation, experimental design, data analysis, or interpretation. The authors take full responsibility for all content.

## A.2  DETAILED RELATED WORK

**Vision-Language Models and Visual Question Answering** Vision-Language Models (VLMs) serve as core technology in multimodal AI, enabling machines to understand and reason about visual content through natural language. Modern VLM development is grounded in contrastive learning, with CLIP (Radford et al., 2021) having pioneered large-scale visual-text alignment. Building on this foundation, BLIP (Li et al., 2022) and BLIP-2 (Li et al., 2023b) established unified frameworks for vision-language understanding and generation, while LLaVA (Liu et al., 2023a) achieved break-throughs in multimodal tasks by integrating vision encoders with large language models.

Recent large VLMs have significantly enhanced visual understanding capabilities. Closed-source models like GPT-4V (Hurst et al., 2024) and Gemini (Comanici et al., 2025) excel at complex visual reasoning, while open-source models such as Qwen2.5-VL (Bai et al., 2025) and InternVL (Chen et al., 2024) provide powerful multimodal tools for the research community.

Specialized techniques have emerged in 3D vision-language understanding (Huang et al., 2025a;b;c; Liu et al., 2025; Song et al., 2025a;b; Ye et al., 2025): Cap3D (Luo et al., 2023) advanced 3D-text data construction through large-scale 3D object descriptions, DiffuRank (Luo et al., 2024) improved caption generation accuracy via optimized rendered view selection, and LLaVA-3D (Zhu et al., 2024) with 3D-LLaVA (Deng et al., 2025) achieved high-quality 3D model descriptions and question-answering.

Traditional 2D visual question answering datasets like VQAv2 (Goyal et al., 2017) have driven visual reasoning development but cannot handle 3D spatial complexity. Early 3D VQA works such as ScanQA (Azuma et al., 2022) focused on indoor spatial relationships, establishing foundations for 3D question answering. However, existing methods show significant limitations when processing cultural artifacts with complex geometric structures and cultural significance. 3D VQA applications in cultural heritage remain underexplored, particularly for ancient Greek pottery with intricate decorative patterns and profound historical meaning. This work constructs the first ancient Greek pottery VaseVQA-3D dataset to address this gap.

**3D Generation and Reconstruction** In image-to-3D model generation, DreamFusion (Poole et al., 2022) pioneered the application of image diffusion priors to 3D generation, proposing the Score Distillation Sampling (SDS) method, which enabled iterative optimization of 3D representations via differentiable volume rendering (Mildenhall et al., 2021). Subsequent studies have made substantial improvements in multiple directions, including 3D representation forms (Lin et al., 2023; Tang et al., 2023a; Yi et al., 2024), sampling strategies (Liang et al., 2024; Wang et al., 2023a;b; Zou et al., 2024), integration of additional geometric cues (Long et al., 2024; Tang et al., 2023b), and consistency in multi-view image generation.

Furthermore, numerous studies have explored training viewpoint-aware image diffusion models based on input images (Chan et al., 2023; Liu et al., 2023c; Shi et al., 2023). A range of research has proposed learning geometric structures in various representation forms from input images through an encoder-decoder network architecture within a deterministic process—such as point clouds (Fan et al., 2017a; Wu et al., 2020), voxels (Girdhar et al., 2016; Wu et al., 2017), meshes (Wang et al., 2018; Worchel et al., 2022), or implicit fields (Mescheder et al., 2019; Xu et al., 2019; Yu et al., 2021).

One-2-3-45 (Liu et al., 2023b) was the first to propose combining a 2D image diffusion model with a multi-view reconstruction model, achieving generative capabilities while maintaining fast reconstruction speed. Recently, some researchers have attempted to train latent 3D diffusion models based on massive high-quality 3D models (Hong et al., 2024; Lan et al., 2024; Li et al., 2024; Wu

et al., 2024; Zhang et al., 2024), demonstrating impressive 3D generation results. However, these methods still have limitations in the task of "high-fidelity generation with image alignment."

TripoSG (Li et al., 2025) adopted a 3D representation with stronger geometric expression ability, improved the diffusion model architecture and training strategies, and achieved state-of-the-art performance in 3D shape generation tasks. Hunyuan3D (Lai et al., 2025) employed a two-stage approach: first, it used a multi-view diffusion model to generate multi-view RGB images, and then converted these images into 3D assets using a Transformer-based large-scale reconstruction model for sparse viewpoints.

**Cultural Heritage and Archaeological AI** Currently, AI technologies are effectively enhancing the level of cultural heritage preservation. ArchaeoScape (Perron et al., 2024) constructed the world's largest archaeological LiDAR dataset, successfully identifying ancient architectural remains under jungle cover through semantic segmentation models (such as U-Net (Ronneberger et al., 2015) and Swin Transformer (Liu et al., 2021)), with accuracy significantly superior to traditional manual interpretation. (Yang et al., 2024) proposed an improved YOLOv5s model (Khanam & Hussain, 2024), combining multispectral data with vegetation indices, identifying 116 moat sites in northeastern Thailand with 100% detection accuracy, representing a vivid example of identification technology and cultural heritage protection. (Feng et al., 2025) proposed an automated mural line drawing generation method combining CLAHE edge enhancement, neural network (MLineNet), and CycleGAN (Chu et al., 2017) denoising. Using Dunhuang murals as test subjects, the restoration results outperformed existing algorithms in detail, clarity, and smoothness metrics, achieving a Q-value of 89.26% . These studies systematically demonstrated the technological breakthroughs and ethical challenges of AI in cultural heritage preservation and archaeological research, providing important references for interdisciplinary collaboration. In the field of cultural heritage 3D reconstruction, Adamopoulos et al. (2020) critically compared 3D digitization techniques to clarify their application boundaries, Jaramillo & Sipiran (2024) proposed a method using diffusion networks to address incomplete data issues, and Pan et al. (2024) realized 3D reconstruction of relief heritage from single old photos, collectively advancing related research.

### A.3 DATASET FEATURES AND COMPOSITION

Our VaseVQA-3D dataset constitutes the first comprehensive evaluation resource specifically designed for 3D ancient Greek pottery visual question answering, filling the gap in existing VQA datasets in the cultural heritage domain. Table 5 shows detailed statistical comparisons from the original Vase dataset to our final VaseVQA-3D dataset. The strict filtering process reduced the dataset from 3,880 vase entries to 664 unique vase entries. Although the data retention rate is only 17.1%, this ensures that the final dataset achieves good standards in archaeological accuracy, image quality, and metadata completeness.

Regarding question type distribution, our dataset maintains a similar balance to the original VaseVQA dataset, with core attribute questions (fabric, technique, shape, overall, dating, decoration) comprising the main proportion, while specialized attribute questions (attribution, provenance) have relatively lower coverage due to the incompleteness of archaeological records. Figure 2 shows typical examples from our VaseVQA-3D dataset, including high-quality 3D vase models, structured question-answer pairs, and cleaned descriptive captions, demonstrating the dataset's comprehensive capabilities in supporting multimodal understanding of ancient Greek pottery.

**GPT-4o Usage for Data Cleaning** The archaeological metadata in the original VaseVQA dataset already possesses strong structural information but contains significant noise in the complete descriptions, making them difficult to use directly for model training. We use GPT-4o to clean this noise and reorganize the metadata into coherent descriptions consistent with standard museum documentation practices.

For example, the original fragmented format:

> "The overall information is: The Vase Number is 14292; The Fabric is ATHENIAN; The Technique is BLACK-FIGURE; The Shape Name is LEKYTHOS; The Provenance is GREECE, ATTICA, MARATHON; The Date is -525 to -475; The Decoration is Body: HERAKLES AND THE BOAR (?); The Collection Record is Athens, National Museum: 1021; ..."

is reorganized into:

> "Athenian black-figure lekythos, c. 525–475 BCE, depicting Herakles and the boar; Marathon, Attica."

This cleaning process consolidates existing archaeological information without introducing new content. Importantly, our evaluation metrics are independent of language style: retrieval accuracy metrics (R@1/R@5/R@10) measure semantic matching independent of stylistic variation; CLIP and FID scores are computed from visual-semantic alignment rather than language patterns; and human evaluation by archaeological experts assesses archaeological accuracy rather than language style. Therefore, the evaluation results reward models that understand archaeological content, not those that merely imitate GPT-4o's language patterns.

Table 5: Dataset Statistics Comparison: Before and After Quality Filtering.

| Metric | Original VaseVQA | Filtered VaseVQA-3D |
|---|---|---|
| Total Vase Entries | 3,880 | 664 |
| Total QA Pairs | 26,101 | 4,460 |
| Avg. QA per Entry | 6.73 | 6.72 |
| Fabric Questions | 3,880 (14.9%) | 664 (14.9%) |
| Technique Questions | 3,880 (14.9%) | 664 (14.9%) |
| Shape Questions | 3,880 (14.9%) | 664 (14.9%) |
| Caption Questions | 3,880 (14.9%) | 664 (14.9%) |
| Dating Questions | 3,872 (14.8%) | 664 (14.9%) |
| Decoration Questions | 3,870 (14.8%) | 663 (14.9%) |
| Attribution Questions | 1,696 (6.5%) | 280 (6.3%) |
| Provenance Questions | 1,143 (4.4%) | 197 (4.4%) |
| Question Types | 8 | 8 |
| Unique Images | 3,880 | 664 |
| Data Retention Rate | 100% | 17.1% |
| Quality Score | Mixed | High |

## A.4 HYPERPARAMETERS AND IMPLEMENTATION DETAILS

**Data Filtering Stage** ResNet-50 binary classifier: cross-entropy loss, Adam optimizer (learning rate 1e-4), 20 epochs, 512×512 pixel RGB input, confidence threshold 0.5. CLIP ViT-B/32 model: fragment removal threshold 0.1, text prompt "a complete intact vase viewed from the front".

**3D Generation Stage** TripoSG: 512×512 pixel front-view input, image preprocessing (normalization, denoising), approximately 5 minutes per sample (single A100), GLB format output. Blender 3.6: 16-frame 360-degree rotation videos, 512×512 pixels per frame, 2fps.

**Post Training Stage** Supervised Fine-tuning (SFT): Qwen2.5VL-3B/7B base models, LoRA rank 8, alpha 32, learning rate 1e-4, batch size 1 (16-step gradient accumulation), 2 epochs , frozen ViT parameters, approximately 2 hours training time (single A100, per model / for both model sizes).

Reinforcement Learning (GRPO): policy gradient methods, policy update every 100 samples, learning rate 1e-5, batch size 8, 10 epochs, approximately 10 hours training time (single A100, per model / for both model sizes).

**Evaluation Metrics** Quantitative: FID Score, CLIP Score, R@1/5/10, lexical similarity. Qualitative: description accuracy, cultural appropriateness (10 archaeological experts).

## A.5 DETAILED WORKFLOW AND IMPLEMENTATION

We adopt a three-stage progressive filtering mechanism to process the original VaseVQA dataset. The initial stage uses a ResNet-50 architecture binary classifier for coarse-grained screening, automatically identifying and removing low-quality images. Building on this, we introduce the CLIP ViT-B/32 model for fine-grained semantic filtering, including fragment removal and optimal view selection.

Table 6: Expert and Non-Expert Evaluation of TripoSG vs Hunyuan3D (Scale: 0-5).

| Dimension | Method | Exp-1 | Exp-2 | Exp-3 | Exp-4 | Exp-5 | Non-1 | Non-2 | Non-3 | Non-4 | Non-5 | Ave. |
|---|---|---|---|---|---|---|---|---|---|---|---|---|
| Geometric Accuracy | TripoSG | 4.5 | 4.6 | 4.2 | 4.4 | 4.5 | 4.3 | 4.1 | 4.4 | 4.2 | 4.3 | 4.35 |
|  | Hunyuan3D | 4.3 | 4.2 | 4.4 | 4.1 | 4.0 | 4.4 | 4.3 | 4.0 | 4.3 | 4.2 | 4.22 |
| Decoration Fidelity | TripoSG | 4.2 | 4.3 | 4.0 | 4.1 | 4.2 | 4.1 | 3.9 | 4.2 | 4.0 | 4.1 | 4.11 |
|  | Hunyuan3D | 4.3 | 4.4 | 4.2 | 4.3 | 4.1 | 4.2 | 4.3 | 4.1 | 4.2 | 4.3 | 4.24 |
| Archaeological Credibility | TripoSG | 4.4 | 4.5 | 4.1 | 4.3 | 4.4 | 4.2 | 4.0 | 4.3 | 4.1 | 4.2 | 4.25 |
|  | Hunyuan3D | 4.1 | 4.0 | 4.2 | 3.9 | 3.8 | 4.2 | 4.1 | 3.8 | 4.1 | 4.0 | 4.02 |
| Overall Average | TripoSG | 4.37 | 4.47 | 4.10 | 4.27 | 4.37 | 4.20 | 3.97 | 4.30 | 4.10 | 4.20 | **4.24** |
|  | Hunyuan3D | 4.23 | 4.20 | 4.27 | 4.10 | 3.97 | 4.27 | 4.23 | 3.97 | 4.20 | 4.17 | 4.16 |

Subsequently, we conduct a systematic comparison between TripoSG and Hunyuan3D based on the VaseEval dataset. After determining TripoSG's advantages in generation quality, we adopt this method for single-image 3D reconstruction of filtered high-quality images. We then convert the generated GLB files to video sequence format using Blender 3.6 for model training.

The supervised fine-tuning stage uses Qwen2.5VL as base models, adopting LoRA for parameter-efficient fine-tuning. Building on SFT, we adopt the GRPO method for verifiable reinforcement learning training. Finally, we conduct comprehensive evaluation of the four trained VaseVLM variants using both quantitative metrics and human evaluation by archaeological experts.

## A.6 Detailed 3D Generation Method Comparison

Table 2 presents a comprehensive comparison of TripoSG and Hunyuan3D across seven dimensions on VaseEval. In terms of traditional image quality metrics, both methods perform closely, but each has advantages. Hunyuan3D has a slight advantage in PSNR (17.23 vs 17.21), while TripoSG performs better in LPIPS (0.1308 vs 0.1319). In SSIM metrics, both perform comparably (TripoSG: 0.8676 vs Hunyuan3D: 0.8657).

In terms of geometric accuracy, the results show differentiation characteristics. TripoSG performs better in Chamfer distance (0.1490 vs 0.1515), showing its advantage in overall geometric reconstruction accuracy. However, Hunyuan3D performs better in normal consistency (0.7389 vs 0.7232), indicating its capability in surface details and lighting interaction.

In semantic consistency evaluation, TripoSG demonstrates significant advantages. In CLIP image similarity, TripoSG (0.8896) slightly outperforms Hunyuan3D (0.8837). More importantly, in CLIP text similarity, TripoSG performs excellently (0.9594 vs 0.9237), exceeding Hunyuan3D by about 3.9%. Although our input is only a single image rather than text descriptions, the CLIP text similarity metric reflects the matching degree between generated 3D models and predefined archaeological description templates, which is crucial for subsequent text generation tasks.

Comprehensive analysis shows that Hunyuan3D has slight advantages in traditional image quality metrics, while TripoSG performs better in geometric reconstruction accuracy and semantic consistency. Considering that our application scenario requires accurate 3D geometric structures and good semantic understanding capabilities to support archaeological description generation, TripoSG's advantages in key metrics make it more suitable for our VQA task requirements.

To further validate the TripoSG selection, we conducted additional blind evaluation with 5 archaeologists and 5 domain-unrelated individuals, assessing both methods on the VaseEval set across three dimensions: geometric accuracy, decoration fidelity, and archaeological credibility.

The human evaluation results confirm TripoSG's superiority, particularly in archaeological credibility (4.25 vs 4.02), which is critical for cultural heritage applications. TripoSG achieves higher overall average (4.24 vs 4.16), validating our method selection.

## A.7 Detailed Dataset Quality Analysis

As shown in Table 3 Among 3D-specialized models, DiffuRank performs best (FID: 0.421, CLIP: 0.798), with this advantage stemming from its specialized architecture design and training strategy for 3D scene understanding. DiffuRank adopts a diffusion model framework that can better capture the complexity and spatial relationships of 3D geometric structures, which is particularly important when processing the three-dimensional forms of ancient Greek vases. Cap3D follows closely (FID:

0.445, CLIP: 0.792), with its advantages based on large-scale 3D-text pair training reflected in semantic understanding, but slightly inferior to DiffuRank in fine-grained control of generation quality. Although LLaVA3D performs relatively weakly in retrieval tasks (R@10: 10.42%), its multimodal fusion mechanism provides important references for subsequent model design.

General-purpose VLMs show significant performance differentiation characteristics. Gemini-2.5-flash performs excellently in retrieval tasks (R@10: 28.57%), benefiting from Google's pretraining advantages on large-scale multimodal data and its advanced attention mechanism design, enabling the model to better establish correspondences between visual features and text descriptions. However, its relative disadvantage in lexical similarity (0.210) reflects the limitations of general models in understanding professional archaeological terminology. This phenomenon is also reflected in other general models, such as GPT-4o achieving only 0.104 in lexical similarity, indicating that while large-scale general pretraining improves overall understanding capabilities, there are still deficiencies in mastering specific domain terminology.

Claude series models show progressive performance characteristics across versions, with Claude-4-sonnet (FID: 0.353) significantly outperforming Claude-3.5-sonnet (FID: 0.455) and Claude-3.7-sonnet (FID: 0.600), reflecting Anthropic's continuous improvements in model architecture optimization and training data quality enhancement. Particularly noteworthy is Claude-4-sonnet's performance in retrieval tasks (R@10: 23.96%) showing significant improvement compared to earlier versions, indicating progress in multimodal understanding and cross-modal retrieval capabilities in new versions.

Among GPT series models, GPT-4.1 performs well in retrieval tasks (R@10: 25.00%), but GPT-4o's relatively lower performance (FID: 0.582) may be related to its design tendency toward dialogue optimization rather than visual understanding tasks. This performance difference reveals the impact of different model optimization objectives: models specifically optimized for visual understanding typically perform better in image-text matching tasks, while dialogue-optimized models may have advantages in generation fluency but are relatively weaker in precise visual understanding.

Notably, open-source models demonstrate competitive capabilities comparable to closed-source commercial models in certain dimensions. Qwen2.5-VL-7B performs excellently in FID metrics (0.334), second only to Gemini-2.5-flash, reflecting Alibaba's technical strength in VLM architecture design. More importantly, this model achieves 0.217 in lexical similarity, significantly outperforming most closed-source models, indicating the potential of the open-source community in specific task optimization. InternVL's strong performance in CLIP scores (0.771) demonstrates its capabilities in semantic understanding, benefiting from its innovative vision-language interaction mechanisms and large-scale pretraining strategies.

The consistent improvement of reinforcement learning training versions compared to supervised fine-tuning versions (7B-RL vs 7B-SFT: FID improvement 1.2%, R@10 improvement 2.0%) demonstrates the effectiveness of the GRPO method in archaeological VQA tasks. This improvement is particularly evident in lexical similarity metrics, with the 7B-RL model achieving 0.276 compared to the SFT version's 0.272, indicating that reinforcement learning training effectively improves the model's mastery of archaeological professional terminology.

Comparing the performance of models with different parameter scales, we observe that larger models generally perform better in most metrics. The performance improvement of VaseVLM-7B compared to VaseVLM-3B (FID: 0.328 vs 0.363) is significant, indicating clear advantages of larger model capacity. In lexical similarity, the 7B-RL model achieves 0.276 compared to the 3B-RL version's 0.245, demonstrating that larger models have superior capabilities in mastering archaeological terminology. However, 3B models still show competitive performance in other metrics, indicating good practical value in resource-constrained scenarios.

Comprehensive analysis shows that specialized training can effectively compensate for disadvantages in model scale. Our VaseVLM-3B-RL (FID: 0.363) outperforms general models with larger parameter scales in multiple metrics, such as its performance in retrieval tasks (R@10: 17.71%) approaching Qwen2.5-VL-7B's 18.75%, demonstrating the advantages of task-specific optimization over pure scale expansion. The evaluation results further reveal inherent characteristic differences of different model architectures when processing 3D visual understanding tasks. Transformer-based models generally perform excellently in semantic understanding, while specialized 3D models have structural advantages in geometric feature capture. Our hybrid training strategy successfully com-

bines the advantages of both aspects, enhancing the perception of 3D geometric structures while maintaining semantic understanding capabilities.

## A.8 ARCHAEOLOGICAL QUALITATIVE ANALYSIS

Beyond quantitative metrics, we conduct qualitative analysis to demonstrate that VaseVLM has genuinely learned specialized archaeological knowledge rather than merely memorizing patterns. We compare VaseVLM-7B-RL with Gemini-2.5-flash across three representative cases.

**Case 1: Red-Figure vs Black-Figure Classification**

> **Ground Truth:** "Athenian Red-Figure Cup, c. 500–450 BCE, depicting a youth wreathing an altar; Detroit Institute of Arts."
>
> **Gemini 2.5 Flash (R@1):** "Athenian black-figure kylix, c. 550-500 BCE, with figural decoration; Attica."
>
> **VaseVLM-7B-RL (R@1):** "Athenian red-figure cup, c. 500–450 BCE, depicting a youth at an altar; Detroit Institute of Arts."

Gemini's R@1 error (Black-Figure vs Red-Figure) represents a fundamental technical classification error, not a minor detail difference. Black-figure and Red-figure techniques represent distinct historical periods separated by 50-100 years. This error is archaeologically unacceptable, as it misleads researchers about chronology and artistic development. VaseVLM correctly identifies the technique, demonstrating learned archaeological knowledge.

**Case 2: Vessel Type Identification (Hydria vs Amphora)**

> **Ground Truth:** "Athenian black-figure hydria, c. 525–475 BCE, depicting Herakles, Dionysos, Hermes, and Athena; Munich Collection."
>
> **Gemini 2.5 Flash (R@1):** "Athenian black-figure amphora, c. 550–500 BCE, depicting mythological scene with multiple figures."
>
> **VaseVLM-7B-RL (R@1):** "Athenian black-figure hydria, c. 525–475 BCE, depicting Herakles, Dionysos, Hermes, and Athena."

The Amphora vs Hydria distinction is critical: these vessels served different functions in ancient Greek society (storage vs water-carrying), leading to different morphological features. Misidentification propagates errors in understanding vessel function, daily life practices, and museum classification. VaseVLM correctly identifies the vessel type and associated mythological figures, indicating genuine archaeological understanding.

**Case 3: Descriptive Style and Generalization**

> **Ground Truth:** "Athenian black-figure amphora, c. 550–500 BCE, depicting figures."
>
> **Gemini 2.5 Flash (R@1):** "Athenian black-figure amphora, c. 550–500 BCE, depicting figures."
>
> **VaseVLM-7B-RL (R@1):** "Athenian black-figure lekythos, c. 500–450 BCE, depicting a woman with a lyre; National Museum, Warsaw."

VaseVLM's answer differs from ground truth, indicating it has developed its own descriptive style emphasizing decorative patterns and professional terminology, rather than memorizing exact answers. This demonstrates genuine archaeological understanding and generalization capability.

**Summary**  These qualitative cases demonstrate that in specialized expert domains like archaeology, R@1 precision is more critical than R@10 recall. Archaeologists require accurate first answers to avoid propagating fundamental errors in research and classification. VaseVLM's superior R@1 performance reflects genuine domain expertise rather than statistical pattern matching.

## A.9 QUALITY COMPARISON ACROSS VLMS

Figure 7 illustrates differences in text generation performance across VLMs on the vase dataset for QA and captioning tasks. Figures 7a and 7b illustrate QA quality comparisons across multiple VLMs, while Figure 7c shows caption generation quality comparison.

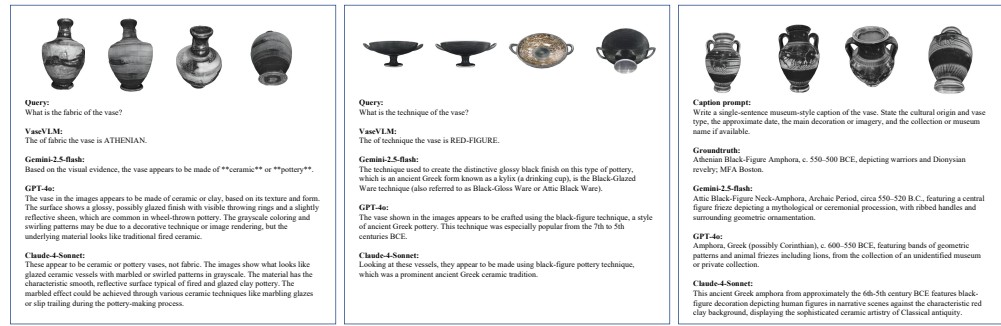

(a) QA quality example.    (b) QA quality example.    (c) Caption quality example.

Figure 7: Examples of QA and captioning quality comparisons across different VLMs. (a) and (b) show QA cases, while (c) figure illustrates a captioning case.

Table 7: Data Processing Results Across Cultural Heritage Domains.

| Domain | Initial | Filtered | 3D Models | Retention | QA Pairs |
|---|---|---|---|---|---|
| Ancient Greek Pottery | 30,000 | 3,880 | 664 | 2.2% | 4,460 |
| Chinese Bronze | 100 | 73 | 52 | 52.0% | 312 |
| Ancient Greek Sculpture | 100 | 63 | 58 | 58.0% | 348 |

Table 8: Domain-Specific RLVR Dimensions.

| Domain | Reward Dimensions (weights) |
|---|---|
| Ancient Greek Pottery | Fabric (0.20), Technique (0.20), Shape (0.15), Dating (0.15), Decoration (0.20), Attribution (0.10) |
| Chinese Bronze | Casting Material (0.18), Technique (0.22), Dating (0.15), Decoration (0.18), Preservation (0.15), Provenance (0.12) |
| Ancient Greek Sculpture | Clay Type (0.30), Dating (0.20), Style (0.15), Decoration (0.20), Excavation Site (0.15) |

Table 9: Performance on New Domains.

| Domain | Method | FID↓ | CLIP↑ | R@10↑ | R@5↑ | R@1↑ | Lex. Sim.↑ |
|---|---|---|---|---|---|---|---|
| Chinese Bronze | Qwen2.5-VL-7B (Baseline) | 0.356 | 0.732 | 16.68% | 8.33% | 3.23% | 0.227 |
| | BronzeVLM-3B-RL | 0.368 | 0.724 | 15.60% | 8.47% | 2.85% | 0.216 |
| | BronzeVLM-7B-RL | **0.324** | **0.752** | **20.83%** | **10.50%** | **3.50%** | **0.274** |
| Greek Sculpture | Qwen2.5-VL-7B (Baseline) | 0.342 | 0.731 | 18.47% | 8.83% | 2.17% | 0.235 |
| | SculptureVLM-3B-RL | 0.356 | 0.696 | 16.67% | 9.37% | 2.13% | 0.228 |
| | SculptureVLM-7B-RL | **0.337** | **0.748** | **19.53%** | **11.25%** | **3.31%** | **0.263** |

## A.10 PIPELINE GENERALIZATION ACROSS CULTURAL HERITAGE DOMAINS

To validate the generalizability of our methodology beyond ancient Greek pottery, we conducted supplementary experiments on Chinese bronze artifacts and ancient Greek sculptures. Both domains share similar characteristics (complex geometric structures, rich decorative patterns, established archaeological classification systems) yet differ in domain-specific knowledge requirements.

**Experimental Setup** For each domain, we customized the RLVR reward dimensions to reflect domain-specific archaeological knowledge while maintaining the core pipeline structure. Table 7 presents data processing results across three domains.

**Domain-Specific Reward Dimensions** Table 8 shows the customized RLVR dimensions for each domain. While maintaining six (five in Greek Sculpture) semantic dimensions, we adjusted weights and focus areas based on archaeological significance.

**Performance Results** Table 9 demonstrates consistent improvements across domains. VaseVLM-7B-RL achieves 8.4-52.5% improvement in R@1 accuracy and 11.9-20.7% improvement in lexical similarity over baseline models.

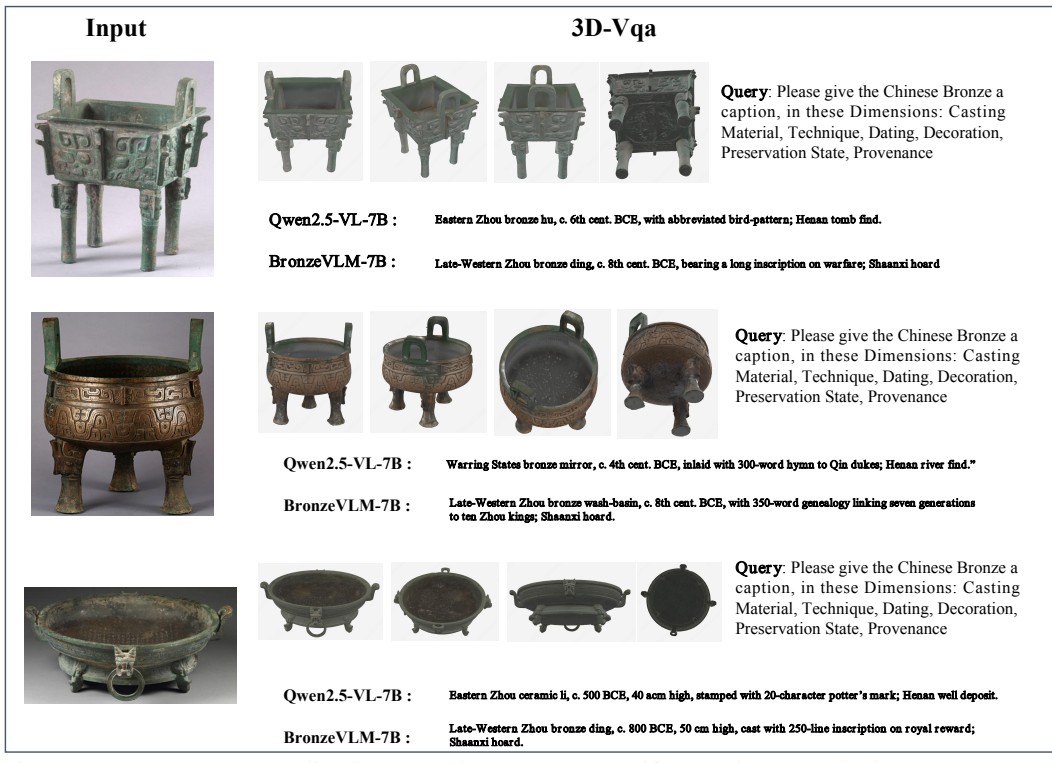

Figure 8: VaseVLM Generalization on Chinese Bronze Artifacts. The example demonstrates successful adaptation to domain-specific archaeological knowledge through customized reward dimensions for casting material and preservation state.

**Qualitative Analysis** Figures 8 and 9 show VQA examples from Chinese bronze artifacts and ancient Greek sculptures, respectively, demonstrating the model's ability to adapt to domain-specific archaeological features.

These results validate the generalizability of our pipeline across diverse cultural heritage domains through flexible reward dimension customization, demonstrating its applicability beyond ancient Greek pottery.

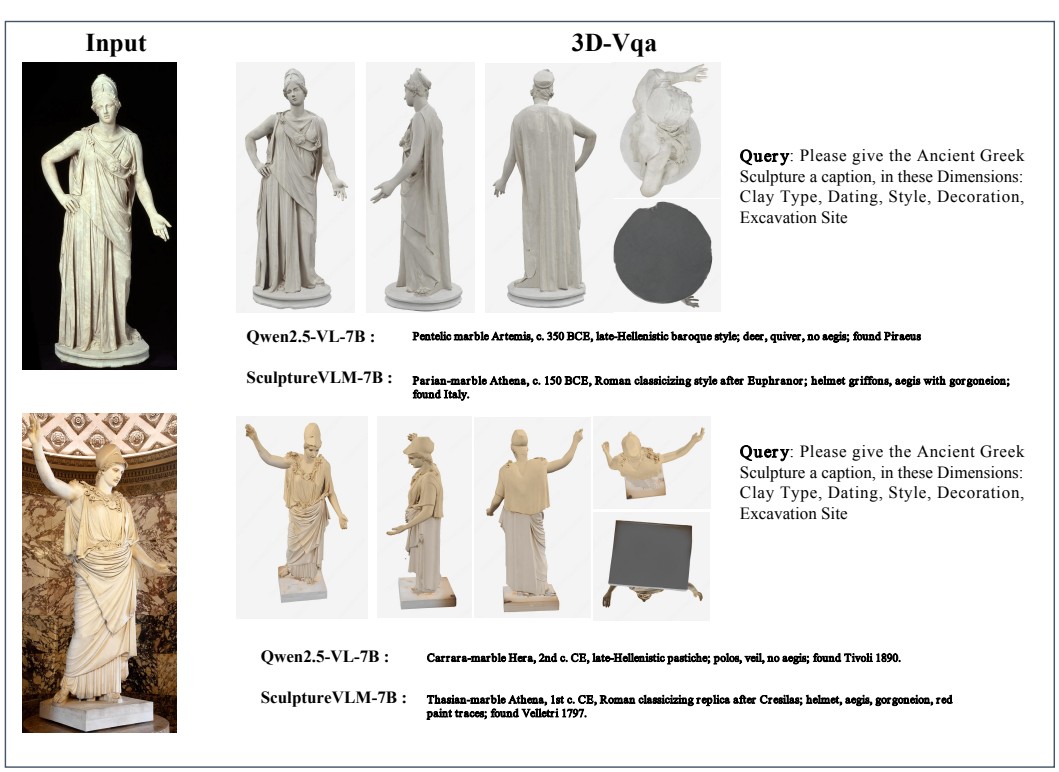

Figure 9: VaseVLM Generalization on Ancient Greek Sculptures. The example demonstrates successful adaptation to domain-specific archaeological knowledge through customized reward dimensions for clay type and excavation site.

