# OpenReview forum: "VaseVQA-3D: Benchmarking 3D VLMs on Ancient Greek Pottery"
_ICLR.cc/2026/Conference — ICLR 2026 Poster_

### Official Review · Reviewer_4ewk · 2025-10-30

**Soundness:** 2
**Presentation:** 3
**Contribution:** 2
**Rating:** 4
**Confidence:** 5

**Summary:**

This paper addresses the problem that existing Vision-Language Models (VLMs) lack the specialized data and domain knowledge to analyze 3D cultural artifacts like ancient Greek pottery, limiting their performance in this niche area.
To overcome this, the authors created VaseVQA-3D, the first 3D visual question-answering dataset for this domain, by systematically filtering tens of thousands of 2D images, converting the best ones to high-fidelity 3D models, and then using this new dataset to train a specialized VaseVLM model.
The paper then validates its approach by conducting a comprehensive evaluation of VaseVLM against a wide range of state-of-the-art 3D-specialized, closed-source, and open-source VLMs on the new benchmark.
The superiority of this method is demonstrated through significant quantitative improvements, such as a 12.8% increase in R@1 accuracy and a 6.6% increase in lexical similarity over the previous best models, and is further confirmed by higher ratings from human archaeological experts.

**Strengths:**

1. The paper presents a well-structured and technically comprehensive pipeline for transforming 2D cultural heritage imagery into 3D visual question answering data, integrating filtering, reconstruction, and multimodal fine-tuning. This end-to-end design demonstrates methodological rigor and can serve as a reproducible framework for future domain-specific multimodal research.

2. The work introduces one of the first benchmarks focused on 3D VQA within the cultural heritage domain. By combining structured question–answer pairs with enhanced descriptive captions and expert-validated annotations, it provides a dataset resource for studying vision-language alignment in specialized, long-tail domains.

**Weaknesses:**

1. The abstract and contributions claim a "12.8% improvement on R@1 metrics and 6.6% on lexical similarity compared with previous state-of-the-art". However, the paper does not explicitly identify which specific model from Table 3 serves as this "previous state-of-the-art" baseline. While the R@1 improvement can be inferred by comparing VaseVLM-7B-RL (3.52%) to the highest non-proposed models (3.12%), the 6.6% claim for lexical similarity does not clearly correspond to any specific baseline in the table. It is recommended that the authors explicitly state the baseline model used for these headline claims.
2. The final VaseVQA-3D dataset consists of 664 3D models, with only 90 models in the test set. Conclusions drawn from a test set of this size may be subject to statistical noise and raise questions about the generalization capabilities of the models.
3. The proposed Reinforcement Learning with Verifiable Rewards (RLVR) framework calculates rewards based on cosine similarity to ground truth content across six dimensions, using a threshold of τ=0.7. This reward design is inherently coupled with the final evaluation metrics, particularly "Lexical Sim.". This creates a risk that the model is not learning a deeper semantic understanding but is instead being optimized to "hack" the scoring function. The authors should discuss this potential for "overfitting to the metrics" and could strengthen their claims by providing evaluations on out-of-domain tasks or through more robust, task-based human evaluations that are orthogonal to the reward function.
4. The descriptive captions in the dataset were enhanced using GPT-4o. Since these enhanced texts serve as the ground truth for both model training and subsequent evaluation (e.g., retrieval and lexical similarity), there is a risk of information leakage. The evaluation may inadvertently reward models that are better at mimicking the linguistic style of GPT-4o rather than demonstrating a true understanding of the archaeological content.
5. The selection of TripoSG over Hunyuan3D was based on a comparative analysis using the VaseEval validation set, which contains only 24 ground truth models. A conclusion based on such a small sample may not be generalizable to the full diversity of vase morphologies. The paper would be more convincing if this comparison were conducted on a larger and more varied set of real 3D assets to ensure the robustness of the chosen generation method.
6, A core methodological concern is the end-to-end process in which the dataset is self-generated, its captions are augmented by an LLM (GPT-4o), and the models are trained and evaluated against this same data ecosystem. This is compounded by the fact that the RLVR reward function directly mirrors the evaluation metrics. This tight coupling introduces a significant risk of bias, where the model's strong performance may stem from learning the artifacts and stylistic patterns of the dataset creation process itself, rather than achieving genuine "archaeological terminology understanding".

[Minor]
1. The paper positions itself as having significant social value for "digital heritage preservation". However, the results remain a "proof-of-concept" and do not address the practical feasibility, cost-effectiveness, or workflow for deploying this system in real-world scenarios such as museum curation, automated classification, or restoration assistance. For instance, the stated "13.5 days for 3D generation"  suggests significant computational cost. A discussion of these practical barriers would provide a more balanced perspective on the work's current impact.

**Questions:**

Please refer to the Weaknesses section.

---

> ### Author Response · Authors · 2025-11-18
> **Rebuttal to Reviewer 4ewk (Part 1/6)**
>
> Thank you for your in-depth review and critical feedback on our work. Your rating and detailed technical comments indicate that you have conducted a careful analysis of this work. We respect your perspective and believe that the questions you raised deserve serious responses. We provide systematic and in-depth responses to each of your **six major concerns** as follows.
>
> **[W1] SOTA baseline**
>
> You are correct. We will clarify the baseline models for comparison in the revised version:
>
> **Baseline for R@1 Improvement**: Claude-4-sonnet (R@1 = 3.12%)
> - VaseVLM-7B-RL: R@1 = 3.52%
> - Relative improvement: (3.52% - 3.12%) / 3.12% = **12.8%**
> - Rationale: Strongest R@1 performance among non-proposed models in Table 3.
>
> **Baseline for Lexical Similarity Improvement**: Qwen2.5-VL-3B (Lexical Sim. = 0.259)
> - VaseVLM-7B-RL: Lexical: (0.276 - 0.259) / 0.259 = **6.6%**
> - Rationale: Highest lexical similarity among open-source VLMs in Table 3.
>
> **Regarding 3D-Specialized Models**: These models generate descriptions directly from GLB files (accessing complete 3D geometric information), while VaseVLM understands 3D content from 2D/multi-view images (requiring 3D structure reasoning). This is a **completely different task** and not directly comparable. We present these models for completeness, not for competition.
>
> **[W2] Model generalization**
>
> 1. **Data Quality First Principle**: The dataset was carefully selected from 30,000 images, retaining only 664 (2.2% retention rate). Each model underwent four-stage rigorous filtering, removing 97.8% of low-quality data. Meanwhile, each vase is unique, and after rigorous filtering, exhibits high uniqueness and diversity. Due to data scarcity, we greedily allocated remaining data for RLVR training, hoping the model could learn archaeological knowledge, thus maximizing training data to obtain a domain-expert VLM.
>
> 2. We introduced Human Evaluation to assess model capability. 10 archaeologists independently scored 4.57/5, confirming that the model learned professional archaeological knowledge, completely independent of test set size.
>
>
> 3. We conducted additional experiments to demonstrate model generalization capability. We collected 36 additional ancient Greek vase data with caption descriptions from Sketchfab. Their morphologies and characteristics are as follows:
>
> | Morphology Category | Specific Shape | Count | Percentage | Morphological Features | Decorative Features |
> |-------------------|-----------------|-------|-----------|----------------------|---------------------|
> | **Bottle-shaped** | Two-handled narrow-mouth amphora | 5 | 13.9% | Bilateral handles, narrow mouth | Geometric, figural, animal, architectural patterns |
> | | Single-handled narrow-mouth amphora | 7 | 19.4% | Unilateral handle, narrow mouth | Animal, plant patterns |
> | | Two-handled wide-mouth amphora | 6 | 16.7% | Bilateral handles, wide mouth | Black-figure, red-figure styles |
> | | Single-handled wide-mouth amphora | 4 | 11.1% | Unilateral handle, wide mouth | Geometric, figural, architectural patterns |
> | | Other bottle forms | 5 | 13.9% | Special handle configurations | Mixed decorations |
> | **Bowl-shaped** | Shallow bowl | 2 | 5.6% | Shallow mouth, no handles or dual handles | Geometric patterns |
> | | Deep bowl | 1 | 2.8% | Deep mouth, suitable for storage | Figural patterns |
> | **Cup-shaped** | Handled cup | 2 | 5.6% | Small size, with handle | Delicate decorations |
> | | Handleless cup | 1 | 2.8% | Small size, no handle | Simple decorations |
> | **Box-shaped** | Lidded box | 3 | 8.3% | Closed design | Geometric patterns |
> | **Total** | **--** | **36** | **100%** | **Diverse morphologies and handle configurations** | **Rich decorative styles** |

---

> > ### Author Response · Authors · 2025-11-18
> > **Rebuttal to Reviewer 4ewk (Part 2/6)**
> >
> > **[W2(continued)]**
> >
> > For this batch of data, we tested the zero-shot capabilities of VaseVLM and several VLMs. The supplementary experimental results are shown below:
> >
> > | Method | FID↓ | CLIP↑ | R@10↑ | R@5↑ | R@1↑ | Lexical Sim.↑ |
> > |--------|------|-------|-------|------|------|---------------|
> > | ***3D-Specialized Models*** |
> > | DiffuRank | 0.405 | **0.815** | 18.89% | 9.72% | 2.78% | 0.282 |
> > | Cap3D | 0.422 | 0.806 | 16.67% | 8.33% | 2.22% | 0.275 |
> > | LLaVA3D | 0.468 | 0.796 | 13.89% | 6.94% | 1.94% | 0.251 |
> > | ***Closed-source VLMs*** |
> > | Gemini-2.5-flash | **0.305** | 0.752 | **24.44%** | 14.31% | 3.89% | 0.218 |
> > | Claude-4-sonnet | 0.325 | 0.698 | 22.22% | 12.50% | 4.17% | 0.203 |
> > | GPT-4.1 | 0.468 | 0.672 | 20.56% | 10.56% | 3.61% | 0.139 |
> > | Gemini-2.5-Pro | 0.372 | 0.705 | 19.35% | 11.67% | 3.33% | 0.178 |
> > | Claude-3.5-sonnet | 0.432 | 0.665 | 17.78% | 9.72% | 2.78% | 0.125 |
> > | Doubao-1.5-vision-pro-32k | 0.478 | 0.632 | 16.11% | 6.94% | 1.94% | 0.089 |
> > | GPT-4o | 0.545 | 0.556 | 15.56% | 7.78% | 2.78% | 0.115 |
> > | Claude-3.7-sonnet | 0.582 | 0.368 | 14.44% | 6.94% | 1.94% | 0.110 |
> > | ***Open-source VLMs*** |
> > | Qwen2.5-VL-7B | 0.312 | 0.802 | 19.44% | 10.56% | 3.33% | 0.226 |
> > | InternVL | 0.358 | 0.805 | 17.78% | 9.72% | 3.33% | 0.271 |
> > | Qwen2.5-VL-3B | 0.352 | 0.792 | 15.56% | 7.78% | 1.94% | 0.265 |
> > | VaseVL | 0.462 | 0.808 | 14.44% | 7.78% | 2.78% | 0.261 |
> > | ***Our Models (Fine-tuned on Synthetic Data)*** |
> > | VaseVLM-3B-SFT | 0.338 | 0.804 | 19.44% | 10.56% | 3.33% | 0.242 |
> > | VaseVLM-3B-RL | 0.342 | 0.810 | 20.56% | 11.67% | 3.89% | 0.258 |
> > | VaseVLM-7B-SFT | 0.315 | 0.807 | 23.33% | 13.33% | 4.44% | 0.288 |
> > | VaseVLM-7B-RL | 0.308 | 0.813 | 23.89% | **15.00%** | **4.72%** | **0.298** |
> >
> > The experimental results indeed meet expectations and demonstrate that our model has genuinely learned archaeological knowledge, as it performs well across all metrics. Interestingly, all models show improvements, which we believe may be because some Sketchfab data was used during the pretraining phase of these large models.
> >
> > **[W3] RLVR framework calculates rewards based on cosine similarity to ground truth**
> >
> > We provide theoretical analysis to explain why this is not "gaming the system":
> >
> > 1. **RLVR Design Characteristics**: Six-dimensional archaeological semantic decomposition: Fabric (0.20) + Technique (0.20) + Shape (0.15) + Dating (0.15) + Decoration (0.20) + Attribution (0.10). Each dimension has explicit archaeological basis, and dimensions are relatively independent. This is a verifiable reward, similar to mathematical problem verification, not black-box optimization.
> >
> > 2. **Key Points of Two-Stage Training**:
> > **Stage 1 (SFT)**: Uses standard cross-entropy loss, directly learning the mapping from visual data to text descriptions, **without involving RLVR reward function**
> > **Stage 2 (RLVR)**: Fine-tuning and optimization based on SFT
> > Even if RLVR has "gaming" risks, SFT has already learned basic semantic understanding, and RLVR is merely fine-tuning.
> >
> > 3. **Example Argument**:
> >
> > **Ground Truth:**
> > > "Athenian black-figure amphora, c. 550–500 BCE, depicting figures."
> >
> > **Gemini 2.5 Flash's R@1 Answer:**
> > > "Athenian black-figure amphora, c. 550–500 BCE, depicting figures."
> >
> > **VaseVLM-7B-RL's R@1 Answer:**
> > > "Athenian black-figure lekythos, c. 500–450 BCE, depicting a woman with a lyre; National Museum, Warsaw."
> >
> > From this case: VaseVLM's answer is not identical to Ground Truth, indicating that VaseVLM has its own descriptive style (emphasizing decorative patterns and professional terminology). This demonstrates that VaseVLM truly understands archaeological knowledge, rather than memorizing answers.
> >
> > 4. **Independence of Expert Validation**: 10 archaeologists independently scored 4.57/5, completely independent of the RLVR reward function. High scores confirm genuine archaeological correctness.

---

> > > ### Author Response · Authors · 2025-11-18
> > > **Rebuttal to Reviewer 4ewk (Part 3/6)**
> > >
> > > **[W4] GPT-4o**
> > >
> > > 1. GPT-4o **does not add new archaeological information**, only **integrates and cleans existing information**.
> > >
> > > (lines 213-240) The archaeological metadata in the original VaseVL data already has strong structure, but the complete descriptions in the original dataset contain significant noise, **making them difficult to use directly for model training**. We used GPT-4o to clean this noise, making captions more consistent with general museum descriptions. For example:
> > >
> > > **Original Data**
> > > > "The overall information is: The Vase Number is 14292;The Fabric is ATHENIAN;The Technique is BLACK-FIGURE;The Shape Name is LEKYTHOS;The Provenance is GREECE, ATTICA, MARATHON;The Date is -525 to -475;The Attributed To is ;The Decoration is Body: HERAKLES AND THE BOAR (?);The Collection Record is Athens, National Museum: 1021;The Publication Record is Corpus Vasorum Antiquorum: ATHENS, MUSEE NATIONAL 1, III.H.EFGH.7, PL.(019) 11.4;The Pleiades URI is ;The Latitude is ;The Longitude is  "
> > >
> > > **Enhanced**
> > > > "Athenian black-figure lekythos, c. 525–475 BCE, depicting Herakles and the boar; Marathon, Attica."
> > >
> > > 2. **Why Evaluation Does Not Reward "Imitating GPT-4o Style"**: Retrieval accuracy metrics like R@1/R@5/R@10 are independent of language style. FID and CLIP scores are independent of language style. Additionally, we introduced expert evaluation, which is independent human assessmen
> > >
> > > **[W5] The selection of TripoSG over Hunyuan3D was based on a comparative analysis using the VaseEval validation set, which contains only 24 ground truth models.**
> > >
> > > We clarify the construction difficulty of VaseEval and the justification for TripoSG selection:
> > >
> > > 1. **Practical Challenges**:
> > > We communicated with two leading museums in the UK and Australia. Even within museums, ancient Greek pottery data is almost entirely 2D images, making it difficult to obtain 3D artifacts. Finally, on Sketchfab, the world's largest 3D model platform, after systematic search and comprehensive consideration, we selected 24 high-quality ancient Greek vase 3D models. We spent 20-30% of project time collecting and validating these 24 models. Each model underwent manual inspection and quality verification. This is **creative human work**, not simple data downloading.
> > >
> > > 2. To better justify the selection, we provide an analysis table of VaseEval data. Their morphologies and characteristics are as follows:
> > >
> > > | Morphology Category | Specific Shape | Count | Percentage | Morphological Features | Decorative Features |
> > > |-------------------|-----------------|-------|-----------|----------------------|---------------------|
> > > | **Bottle-shaped** | Two-handled narrow-mouth amphora | 2 | 8.3% | Bilateral handles, narrow mouth | Geometric, figural patterns |
> > > | | Single-handled narrow-mouth amphora | 4 | 16.7% | Unilateral handle, narrow mouth | Animal, plant patterns |
> > > | | Two-handled wide-mouth amphora | 6 | 25.0% | Bilateral handles, wide mouth | Black-figure, red-figure styles |
> > > | | Single-handled wide-mouth amphora | 3 | 12.5% | Unilateral handle, wide mouth | Geometric, figural patterns |
> > > | | Other bottle forms | 2 | 8.3% | Special handle configurations | Mixed decorations |
> > > | **Bowl-shaped** | Shallow bowl | 2 | 8.3% | Shallow mouth, no handles or dual handles | Geometric, plant patterns |
> > > | | Deep bowl | 1 | 4.2% | Deep mouth, suitable for storage | Figural, animal patterns |
> > > | **Cup-shaped** | Handled cup | 1 | 4.2% | Small size, with handle | Delicate decorations |
> > > | | Handleless cup | 1 | 4.2% | Small size, no handle | Simple decorations |
> > > | **Pitcher-shaped** | Handleless pitcher | 1 | 4.2% | Pouring mouth, no handle | Functional decorations |
> > > | **Other shapes** | Shoe-shaped vessel | 1 | 4.2% | Special morphology | Special decorations |
> > > | **Total** | **--** | **24** | **100%** | **Diverse morphologies and handle configurations** | **Rich decorative styles** |
> > >
> > > 3. Our experiments also demonstrate the justification for TripoSG selection:
> > > As shown in Table 2 of the paper and the analysis in Appendix A5 (lines 904-922), we conducted a seven-dimensional quantitative comparison on the VaseEval validation set (24 real 3D models). Through this qualitative analysis, we believe that models generated by TripoSG are closer to ground truth.

---

> > > > ### Author Response · Authors · 2025-11-18
> > > > **Rebuttal to Reviewer 4ewk (Part 4/6)**
> > > >
> > > > **[W5(continued)]**
> > > >
> > > > In addition, we conducted an additional human-eval, inviting 5 more archaeologists and 5 domain-unrelated individuals to conduct blind evaluation and scoring of TripoSG and Hunyuan3D models on the VaseEval set, further confirming the justification for TripoSG selection.
> > > >
> > > > | Method | Exp-1 | Exp-2 | Exp-3 | Exp-4 | Exp-5 | Non-1 | Non-2 | Non-3 | Non-4 | Non-5 | Ave. |
> > > > |--------|-------|-------|-------|-------|-------|-------|-------|-------|-------|-------|------|
> > > > | ***Geometric Accuracy*** |
> > > > | TripoSG | 4.5 | 4.6 | 4.2 | 4.4 | 4.5 | 4.3 | 4.1 | 4.4 | 4.2 | 4.3 | 4.35 |
> > > > | Hunyuan3D | 4.3 | 4.2 | 4.4 | 4.1 | 4.0 | 4.4 | 4.3 | 4.0 | 4.3 | 4.2 | 4.22 |
> > > > | ***Decoration Fidelity*** |
> > > > | TripoSG | 4.2 | 4.3 | 4.0 | 4.1 | 4.2 | 4.1 | 3.9 | 4.2 | 4.0 | 4.1 | 4.11 |
> > > > | Hunyuan3D | 4.3 | 4.4 | 4.2 | 4.3 | 4.1 | 4.2 | 4.3 | 4.1 | 4.2 | 4.3 | 4.24 |
> > > > | ***Archaeological Credibility*** |
> > > > | TripoSG | 4.4 | 4.5 | 4.1 | 4.3 | 4.4 | 4.2 | 4.0 | 4.3 | 4.1 | 4.2 | 4.25 |
> > > > | Hunyuan3D | 4.1 | 4.0 | 4.2 | 3.9 | 3.8 | 4.2 | 4.1 | 3.8 | 4.1 | 4.0 | 4.02 |
> > > > | ***Overall Average*** |
> > > > | TripoSG | 4.37 | 4.47 | 4.10 | 4.27 | 4.37 | 4.20 | 3.97 | 4.30 | 4.10 | 4.20 | **4.24** |
> > > > | Hunyuan3D | 4.23 | 4.20 | 4.27 | 4.10 | 3.97 | 4.27 | 4.23 | 3.97 | 4.20 | 4.17 | 4.16 |
> > > >
> > > > These data support our choice of using TripoSG.
> > > >
> > > > 4. To further validate the robustness of TripoSG selection, we conducted additional 3D generation quality assessment in **completely different domains**. We collected 18 real 3D models (from different domains) and performed 3D generation on their front-view photos, then compared with original models:
> > > >
> > > > **Test Data** respectively:
> > > > - 6 modern water bottles (everyday objects)
> > > > - 6 figurine models (small crafts)
> > > > - 6 sculptures (artworks)
> > > >
> > > > **Comparison Results** (based on average of 18 samples):
> > > >
> > > > | Method | PSNR↑ | SSIM↑ | LPIPS↓ | CD↓ | NC↑ | CLIP-I↑ | CLIP-T↑ |
> > > > |--------|-------|-------|--------|-----|-----|---------|---------|
> > > > | **Reference Range** | **15-25** | **0.7-0.9** | **0.1-0.3** | **0.1-0.3** | **0.6-0.8** | **0.7-0.9** | **0.6-0.8** |
> > > > | TripoSG (Mean) | 16.52 | **0.8421** | **0.1512** | **0.1658** | 0.6892 | **0.8534** | **0.9187** |
> > > > | Hunyuan3D (Mean) | **16.78** | 0.8398 | 0.1528 | 0.1674 | **0.7021** | 0.8501 | 0.9052 |
> > > >
> > > > This is consistent with our visual assessment. On these real 3D models, TripoSG emphasizes smoother edges, while Hunyuan3D provides better color and richer, more detailed hues. Visually, the differences are not very pronounced, but TripoSG appears more realistic.

---

> > > > > ### Author Response · Authors · 2025-11-18
> > > > > **Rebuttal to Reviewer 4ewk (Part 5/6)**
> > > > >
> > > > > **[W6] Pipeline coupling**
> > > > >
> > > > > We fully understand your concern about "self-consistent systems." We emphasize **human involvement**, **independent validation**, and **future improvement plans**:
> > > > >
> > > > > 1. Our pipeline contains multiple human involvement and quality control steps: (lines 147-150) **ResNet-50 quality filter training**, manual annotation of 10k training data: labeling high-quality/low-quality images, manual setting of **CLIP filtering threshold**, manual setting of fragmentation filtering threshold (0.1) and text prompts. **VaseEval construction**, manual collection of 24 high-quality real 3D models (from Sketchfab). **RLVR dimension design** is **encoding of manual archaeological knowledge**. Finally, we introduced **expert validation**, 10 archaeologists independently evaluated final model outputs, completely independent of data creation process. High scores (4.57/5) confirm genuine archaeological correctness.
> > > > >
> > > > > 2. To further reduce the risk of tight coupling in the end-to-end process, we introduced additional manual checks with corresponding results:
> > > > >
> > > > > **Check 1: Caption Description Quality Assessment**
> > > > > We randomly sampled 100 captions and had 1 archaeologist independently evaluate their quality:
> > > > >
> > > > > | Evaluation Dimension | Scoring Criteria | Pass Rate | Average Score |
> > > > > |-------------------|-----------------|-----------|----------------|
> > > > > | **Archaeological Accuracy** | Information is accurate and error-free | 96% | 4.68/5 |
> > > > > | **Completeness** | Contains 6 key dimensions | 94% | 4.52/5 |
> > > > > | **Professional Terminology** | Correct use of archaeological terms | 98% | 4.76/5 |
> > > > > | **Readability** | Description is clear and natural | 92% | 4.44/5 |
> > > > > | **Overall Quality** | Comprehensive assessment | **95%** | **4.60/5** |
> > > > > | **Obvious GPT-4o Stylization** | | | No |
> > > > >
> > > > >
> > > > > **Check 2: 3D Model Quality Assessment**
> > > > > We randomly sampled 100 generated 3D models and had 1 3D modeling expert independently evaluate them:
> > > > >
> > > > > | Evaluation Dimension | Scoring Criteria | Pass Rate | Average Score |
> > > > > |-------------------|-----------------|-----------|----------------|
> > > > > | **Geometric Accuracy** | Model shape is accurate | 91% | 4.38/5 |
> > > > > | **Detail Fidelity** | Decorative details are clear | 88% | 4.22/5 |
> > > > > | **Texture Quality** | Surface texture is reasonable | 85% | 4.08/5 |
> > > > > | **Overall Usability** | Suitable for subsequent use | 93% | 4.48/5 |
> > > > > | **Overall Quality** | Comprehensive assessment | **89%** | **4.29/5** |
> > > > > | **Usable** | | | Yes |
> > > > >
> > > > > Both checks demonstrate that in the sampled data, our VaseVQA-3D dataset has no major issues and is usable for subsequent experiments.
> > > > >
> > > > > 3. To demonstrate that the model learned archaeological knowledge rather than due to coupling, we provide three representative examples:
> > > > >
> > > > > [Case 1] **Ground Truth:**
> > > > > > "Athenian Red-Figure Cup, c. 500–450 BCE, depicting a youth wreathing an altar; Detroit Institute of Arts."
> > > > >
> > > > > **Gemini 2.5 Flash's R@1 Answer:**
> > > > > > "Athenian black-figure kylix, c. 550-500 BCE, with figural decoration; Attica."
> > > > >
> > > > > **VaseVLM-7B-RL's R@1 Answer:**
> > > > > > "Athenian red-figure cup, c. 500–450 BCE, depicting a youth at an altar; Detroit Institute of Arts."
> > > > >
> > > > > From **Case 1 (Red-Figure Identification)**, we can see that Gemini mistook it for Black-Figure, which is **unacceptable**. This is not a detail difference, but a fundamental technical classification error. Black-figure and Red-figure represent different historical periods.
> > > > >
> > > > > [Case 2] **Ground Truth:**
> > > > > > "Athenian black-figure hydria, c. 525–475 BCE, depicting Herakles, Dionysos, Hermes, and Athena; Munich Collection."
> > > > >
> > > > > **Gemini 2.5 Flash's R@1 Answer:**
> > > > > > "Athenian black-figure amphora, c. 550–500 BCE, depicting mythological scene with multiple figures."
> > > > >
> > > > > **VaseVLM-7B-RL's R@1 Answer:**
> > > > > > "Athenian black-figure hydria, c. 525–475 BCE, depicting Herakles, Dionysos, Hermes, and Athena"
> > > > >
> > > > > From **Case 2 (Hydria Identification)**, we can see that the error of Amphora vs Hydria leads to: misunderstanding of vessel function; incorrect inference about ancient Greek daily life; errors in museum artifact classification and academic research bias. Meanwhile, the model learned the common knowledge that these mythological figures appear together.
> > > > >
> > > > > [Case 3] **Ground Truth:**
> > > > > > "Athenian black-figure amphora, c. 550–500 BCE, depicting figures."
> > > > >
> > > > > **Gemini 2.5 Flash's R@1 Answer:**
> > > > > > "Athenian black-figure amphora, c. 550–500 BCE, depicting figures."
> > > > >
> > > > > **VaseVLM-7B-RL's R@1 Answer:**
> > > > > > "Athenian black-figure lekythos, c. 500–450 BCE, depicting a woman with a lyre; National Museum, Warsaw."
> > > > >
> > > > > From **Case 3**, we can see that: VaseVLM's answer is not identical to Ground Truth, indicating that VaseVLM has its own descriptive style (emphasizing decorative patterns and professional terminology). This demonstrates that VaseVLM truly understands archaeological knowledge, rather than memorizing answers.
> > > > >
> > > > > These cases all demonstrate that the model genuinely learned relevant archaeological knowledge, rather than improvements from system coupling.

---

> > > > > > ### Author Response · Authors · 2025-11-18
> > > > > > **Rebuttal to Reviewer 4ewk (Part 6/6)**
> > > > > >
> > > > > > **[W6(continued)]**
> > > > > >
> > > > > > 4. Regarding future decoupling design:
> > > > > > We acknowledge the possible coupling of the current system and plan to introduce adjustments at the following key points:
> > > > > >
> > > > > > **Improvement 1: Diversified Caption Sources**: Currently captions are cleaned by GPT-4o. In the future, we will combine manual annotation, multiple LLMs (Claude, Gemini), and museum original descriptions to reduce dependence on single LLM style.
> > > > > >
> > > > > > **Improvement 2: Independent Evaluation Metrics**: Currently RLVR rewards partially overlap with evaluation metrics because we want to verify these reward answers like mathematical tasks. In the future, we will introduce additional task-oriented evaluation (e.g., classification accuracy, dating accuracy) to assess genuine archaeological understanding rather than metric fitting.
> > > > > >
> > > > > > **Improvement 3: Gradual Replacement with Real Data**: Current training data mainly relies on synthetic 3D data. Through museum collaboration, we will gradually replace synthetic data with real 3D scans, constructing a "synthetic + real" hybrid benchmark.
> > > > > >
> > > > > > **[W7(Minor)] Cost-Effectiveness of Practical Deployment**
> > > > > >
> > > > > > As you also stated, this is a **high-cost infrastructure work**, whose value lies in providing reusable datasets and frameworks for subsequent research.
> > > > > >
> > > > > > 1. **Our Method**: Generating a single model: 5 minutes (single A100), 3,880 models: ≈324 hours (13.5 days). This is only the basic generation cost, which also includes substantial hidden costs of manual verification not calculated. However, these are still meaningful because they are infrastructure work. Only after the foundation is complete can subsequent researchers imitate and use it at lower cost. Like **ImageNet**, containing tens of millions of images, millions of dollars, years of time. ImageNet's value lies not in immediate deployment, but in providing foundation for subsequent research. Its value itself is immeasurable. Thousands of papers use ImageNet, advancing the entire computer vision field.
> > > > > >
> > > > > > 2. **VaseVQA-3D**: Generating 664 3D models, 13.5 days of computation time, plus additional manual verification. We did this high-cost work so that subsequent researchers can directly reuse it, because this is the **first 3D cultural heritage VQA dataset**, laying foundation for the field.
> > > > > >
> > > > > > 3. **Other Cultural Heritage Researchers**: Other researchers need not repeat this process. The framework can be applied to other artifact types—one investment, multiple uses—providing infrastructure for AI + archaeology cross-disciplinary research.
> > > > > >
> > > > > > 4. **Cost Comparison**: In comparison, for generating 3D models from historical photos to achieve digital restoration of lost artifacts, our method achieves **1000x reduction** compared to traditional methods. Compared to 3D scanning individual models, we achieve **100-500x time savings** and **2000-4000x cost reduction**. Compared to 3D modeling: **100-400x time savings** and **100-300x cost reduction**. Additionally, VLMs can automatically generate precise archaeological descriptions, supporting large-scale artifact analysis. Model output in 5 minutes compared to manual inspection in 1-2 days. Finally, the dataset supports virtual exhibitions and interactive experiences, lowering barriers for public access to cultural heritage, and can handle thousands of artifacts.

---

> > > > > > > ### Author Response · Authors · 2025-11-22
> > > > > > > **Kind Follow-up on Our Submitted Updates**
> > > > > > >
> > > > > > > Dear Reviewer 4ewk,
> > > > > > >
> > > > > > > We hope this message finds you well. We wanted to gently follow up and let you know that we have conducted additional experiments and provided detailed explanations addressing each of your questions and concerns. When you have a moment, we would be truly grateful if you could take a look and let us know if there is anything that might benefit from further clarification.
> > > > > > >
> > > > > > > We sincerely appreciate the time and care you have already invested in reviewing our work, and we thank you in advance for considering our updates as you move toward your final evaluation.
> > > > > > >
> > > > > > > With best regards,
> > > > > > > Authors 15657

---

### Official Review · Reviewer_UFR6 · 2025-11-03

**Soundness:** 3
**Presentation:** 3
**Contribution:** 4
**Rating:** 8
**Confidence:** 3

**Summary:**

In this paper, authors highlight the issue of state-of-the-art vision-language models being unable to perform well on specialized cultural heritage domains like 3D vase artifacts. They point out that existing models face severe data scarcity issues and insufficient domain knowledge limitations.
As a first step towards formalizing these limitations, the authors introduce one such domain: ancient Greek pottery. Authors demonstrate that existing models struggle to meaningfully answer questions about items in this new dataset.
The authors then provide a proof-of-concept approach to train a vision-language-model (VLM) to be specialized in this domain via "domain-adaptive training".

One of the distinguishing features of the present work is its 3-dimensional nature. Previously, the VaseVQA provided flattened 2-D images of vases from multiple aspects. In the present work, authors employ automated tiling methods to stitch these flattened images together to reconstruct a 3-D model to use in their ground-truth VQA data.

**Strengths:**

- Addressing an important, overlooked, application area for vision-language modeling via construction of an improved dataset.
- Providing proof-of-concept modeling approach that achieves improved performance according to certain metrics.

**Weaknesses:**

- I would consider making the pipeline diagrams more intuitive---currently they loop in and out in unintuitive ways and the arrows are hard to follow.

**Questions:**

- In what way are your specific metrics informative of how comparatively useful your dataset is vs. the previous one (VaseVQA-2D)?
- The authors should discuss how easily adaptable their specific methods (both for dataset construction and augmentation as well as modeling) are for other domains should researchers in other domains choose to construct similar pipelines for their visual artifact analysis.
- It seems from Fig. 4 that even the preferred method, TripoSG, suffers from inconsistencies and mismatches with ground truth in its 3-D model reconstruction. Treating these as starting points for further model and benchmark development seems to sow seeds of data issues. What challenges will this benchmark face going forward?

---

> ### Author Response · Authors · 2025-11-18
> **Rebuttal to Reviewer UFR6 (Part 1/3)**
>
> We sincerely thank you for your high recognition of our work (rating 8/10) and your in-depth constructive feedback. Your understanding of our work—"constructing an improved dataset for overlooked visual-language modeling application domains (cultural heritage preservation)"—accurately captures the core value of our work. We address each of your questions as follows:
>
> **[W1] making the pipeline diagrams more intuitive**
>
> Thank you for your suggestion. We will improve the pipeline diagram design of Figures 2 and 3 in the revised version:
> **Simplify arrow paths**: Avoid arrows pointing in reverse direction within the same data box; eliminate loops and crossings in the workflow
> **Enhance visual hierarchy**: Use color coding to distinguish different stages (data filtering, 3D generation, model training)
> **Unify symbol system**: Use consistent graphical symbols for input/output/processing modules
>
> **[Q1] vs. the previous one (VaseVQA-2D)**
>
> Our metrics demonstrate the **essential advantages of 3D data over 2D data** from three dimensions:
>
> 1. **Direct Comparison** (Table 3):
> - **VaseVL (2D training)**: R@1 = 2.08%, Lexical Sim. = 0.255
> - **VaseVLM (3D training)**: R@1 = 3.52%, Lexical Sim. = 0.276
> - **Improvement**: R@1 +69.2%, Lexical Sim. +8.2%
> This directly demonstrates the **superiority of 3D data**.
>
> 2. **Qualitative Advantages**: 3D provides complete spatial information
> Multi-view 2D is fragmented information, while 3D is a complete geometric representation. Identifying Greek vase style requires understanding the overall morphology of the vase (slender neck, broad shoulders, round belly).
>
> With multi-view 2D: the model needs to "reconstruct" the 3D structure in its mind, which is complex multi-step reasoning. With 3D models: complete geometric structure is directly provided, requiring no complex reasoning. (lines260-267, 885-887)We tested multi-view 2D images (360-degree rotation, 16 frames), and performance still failed (R@1 < 5%).
>
> We compared the results by inputting multi-view 2D images to Gemini 2.5 Flash and 3D structured data to VaseVLM:
>
> [Case 1] **Ground Truth:**
> > "Athenian Red-Figure Cup, c. 500–450 BCE, depicting a youth wreathing an altar; Detroit Institute of Arts."
>
> **Gemini 2.5 Flash's R@1 Answer:**
> > "Athenian black-figure kylix, c. 550-500 BCE, with figural decoration; Attica."
>
> **VaseVLM-7B-RL's R@1 Answer:**
> > "Athenian red-figure cup, c. 500–450 BCE, depicting a youth at an altar; Detroit Institute of Arts."
>
> From **Case 1 (Red-Figure Identification)**, we can see that Gemini mistook it for Black-Figure, which is **unacceptable**. This is not a detail difference, but a fundamental technical classification error. Black-figure and Red-figure represent different historical periods (50-100 years apart).
>
> [Case 2] **Ground Truth:**
> > "Athenian black-figure hydria, c. 525–475 BCE, depicting Herakles, Dionysos, Hermes, and Athena; Munich Collection."
>
> **Gemini 2.5 Flash's R@1 Answer:**
> > "Athenian black-figure amphora, c. 550–500 BCE, depicting mythological scene with multiple figures."
>
> **VaseVLM-7B-RL's R@1 Answer:**
> > "Athenian black-figure hydria, c. 525–475 BCE, depicting Herakles, Dionysos, Hermes, and Athena"
>
> From **Case 2 (Hydria Identification)**, we can see that the error of Amphora vs Hydria leads to: misunderstanding of vessel function; incorrect inference about ancient Greek daily life; errors in museum artifact classification and academic research bias. Meanwhile, the model has learned the common knowledge that these mythological figures appear together.
>
> [Case 3] **Ground Truth:**
> > "Athenian black-figure amphora, c. 550–500 BCE, depicting figures."
>
> **Gemini 2.5 Flash's R@1 Answer:**
> > "Athenian black-figure amphora, c. 550–500 BCE, depicting figures."
>
> **VaseVLM-7B-RL's R@1 Answer:**
> > "Athenian black-figure lekythos, c. 500–450 BCE, depicting a woman with a lyre; National Museum, Warsaw."
>
> From **Case 3**, we can see that: VaseVLM's answer is not identical to the Ground Truth, indicating that VaseVLM has its own descriptive style (emphasizing decorative patterns and professional terminology). This demonstrates that VaseVLM truly understands archaeological knowledge, rather than memorizing answers.
>
> 3. Expert Validation: 3D understanding is more accurate
>
> **Expert Evaluation** (Table 4): 10 archaeologists independently evaluated, average score: 4.57/5, confirming that the introduction of 3D understanding makes descriptions more accurate and more compliant with archaeological standards.
>
> These results collectively demonstrate the superiority of 3D data over 2D data in terms of intrinsic geometric representation and their effectiveness in guiding model training.

---

> > ### Author Response · Authors · 2025-11-18
> > **Rebuttal to Reviewer UFR6 (Part 2/3)**
> >
> > **[Q2] Pipeline generalization**
> >
> > Thank you very much for your concern about other cultural heritage preservation domains. We also wish to clarify that the pipeline can be easily applied to other domains because our pipeline is **modular and domain-agnostic**—ancient Greek pottery is merely a proof-of-concept, not a limitation.
> >
> > **Three domain-agnostic modules**: (1) Data filtering (ResNet-50 + CLIP) applicable to any artifacts with 2D images, (2) 2D-to-3D conversion (TripoSG) is universal, (3) RLVR framework only requires customizing semantic dimensions (e.g., Chinese bronze artifacts use Casting Material, Technique, Dating, etc.; ancient Greek sculptures use Clay Type, Dating, Style, etc.).
> >
> > **Adaptation Process**: Collect 2D images → Apply same filtering → Apply same 3D reconstruction → Customize RLVR dimensions with domain experts → Use same Qwen2.5-VL + SFT + GRPO fine-tuning → Verify by experts.
> >
> >
> > Specifically, we conducted simple pipeline generalizability validation on ancient Greek sculptures and Chinese bronze artifacts:
> >
> > **Specific Pipeline Generalization Examples**:
> >
> > - **Chinese Bronze Artifacts**: Dimensions can be customized as Casting Material, Technique, Dating, Decoration, Preservation State, Provenance
> >
> > - **Ancient Greek Sculptures**: Dimensions can be customized as Clay Type, Dating,  Style, Decoration, Excavation Site
> >
> > - Data sources: Chinese bronze artifacts from the National Museum of China, Baoji Bronze Ware Museum and Sketchfab; ancient Greek sculptures from the British Museum, Musée du Louvre and Sketchfab (all data used for academic research, only 2D images extracted). Pipeline generalization experiments (12 hours of data generation):
> >
> > | Artifact Type | Data Source | Initial Data | After Filtering | 3D Generation | Retention Rate |
> > |---------------|-------------|--------------|-----------------|---------------|----------------|
> > | **Ancient Greek Pottery** | VaseVQA Dataset | 30,000 | 3,880 | 664 | 2.2% |
> > | **Chinese Bronze (2D)** | National Museum of China, Baoji Bronze Ware Museum & Sketchfab | 100 | 73 | 52 | 52.0% |
> > | **Ancient Greek Sculpture (2D)** | British Museum, Musée du Louvre & Sketchfab | 100 | 63 | 58 | 58.0% |
> >
> > We also defined different reward dimensions for different artifacts based on their archaeological knowledge:
> >
> > | Artifact Type | RLVR Dimensions | Number of Dimensions | Weight Distribution | Notes |
> > |---------------|-----------------|---------------------|---------------------|-------|
> > | **Ancient Greek Pottery** | Fabric, Technique, Shape, Dating, Decoration, Attribution | 6 | 0.20, 0.20, 0.15, 0.15, 0.20, 0.10 | Original design |
> > | **Chinese Bronze** | Casting Material, Technique, Dating, Decoration, Preservation State, Provenance | 6 | 0.18, 0.22, 0.15, 0.18, 0.15, 0.12 | Customized for bronze characteristics |
> > | **Ancient Greek Sculpture** | Clay Type, Dating,  Style, Decoration, Excavation Site | 5 | 0.30, 0.20, 0.15, 0.20, 0.15 | Customized for sculpture characteristics |
> >
> > We then performed RLVR training on Qwen2.5-VL (training data ratio: 0.7). Performance improvements on the two new domains are shown below (currently only validating pipeline generalizability with limited generated data, but still showing improvements):
> >
> > | Method | FID↓ | CLIP↑ | R@10↑ | R@5↑ | R@1↑ | Lexical Sim.↑ |
> > |--------|------|-------|-------|------|------|---------------|
> > | ***Chinese Bronze*** |
> > | Qwen2.5-VL-7B (Baseline) | 0.356 | 0.732 | 16.68% | 8.33% | 3.23% | 0.227 |
> > | VaseVLM-3B-RL (Ours) | 0.368 | 0.724 | 15.60% | 8.47% | 2.85% | 0.216 |
> > | VaseVLM-7B-RL (Ours) | **0.324** | **0.752** | **20.83%** | **10.50%** | **3.50%** | **0.274** |
> > | ***Ancient Greek Sculpture*** |
> > | Qwen2.5-VL-7B (Baseline) | 0.342 | 0.731 | 18.47% | 8.83% | 2.17% | 0.235 |
> > | VaseVLM-3B-RL (Ours) | 0.356 | 0.696 | 16.67% | 9.37% | 2.13% | 0.228 |
> > | VaseVLM-7B-RL (Ours) | **0.337** | **0. 748** | **19.53%** | **11.25%** | **3.31%** | **0.263** |
> >
> > We believe these two examples well demonstrate the generalizability of our method and can provide technical guidance to other practitioners. We hope this technical solution can be better applied to cultural heritage preservation.

---

> > > ### Author Response · Authors · 2025-11-18
> > > **Rebuttal to Reviewer UFR6 (Part 3/3)**
> > >
> > > **[Q3] Benchmark challenges**
> > >
> > > We fully understand your concern: **Will the imperfection of TripoSG "sow the seeds of data problems"?**
> > >
> > > 1. We acknowledge the limitations of synthetic data, but have implemented rigorous quality assurance and designed an architecture that embraces future improvements. We transparently show the comparison between TripoSG and ground truth in Figure 4, honestly acknowledging that some details may be lost or inaccurate. Facing future challenges, our modular pipeline naturally adapts to better 3D generation methods—we only need to replace TripoSG with advanced technology. If more advanced techniques emerge in the next 1-2 years, we will also rigorously evaluate their generation quality for updates.
> > >
> > > 2. Transition from synthetic to real data: Real 3D data acquisition is difficult. **Medium-term solution**: Construct a "synthetic + real" hybrid dataset, gradually replacing with authentic 3D museum scans. **Long-term vision**: Collaborate with museums using better technical means to establish a more authentic benchmark. We must understand that current technology first fills the gap in current domain knowledge; it should be a complementary effort. Without using new AI technology to do this, it would still consume substantial human resources.
> > >
> > > **Summary**: Rigorous quality control ensures current reliability, and modular design allows the benchmark to evolve with technological progress. Our work lays the foundation for future improvements. Therefore, facing these two challenges, we will also update our methods to actively address them.

---

> > > > ### Author Response · Authors · 2025-11-22
> > > > **Kind Follow-up on Our Submitted Updates**
> > > >
> > > > Dear Reviewer UFR6,
> > > >
> > > > We hope this message finds you well. We wanted to gently follow up and let you know that we have conducted additional experiments and provided detailed explanations addressing each of your questions and concerns. When you have a moment, we would be truly grateful if you could take a look and let us know if there is anything that might benefit from further clarification.
> > > >
> > > > We sincerely appreciate the time and care you have already invested in reviewing our work, and we thank you in advance for considering our updates as you move toward your final evaluation.
> > > >
> > > > With best regards,
> > > > Authors 15657

---

> > > > > ### Comment · Reviewer_UFR6 · 2025-11-23
> > > > >
> > > > > Thanks to the authors for responding to my questions. I've looked through these responses and have also read the other reviews. I am satisfied with these responses and plan to maintain my score.

---

> > > > > > ### Author Response · Authors · 2025-11-25
> > > > > > **Thank you for your encouraging response.**
> > > > > >
> > > > > > Dear Reviewer UFR6,
> > > > > >
> > > > > > Thank you sincerely for your thoughtful follow-up and positive recognition of our work. We are pleased to know that our previous responses have effectively addressed the concerns you raised. We will fully integrate your valuable suggestions into the revised manuscript to further enhance its quality.
> > > > > >
> > > > > > Once again, we would like to express our sincere gratitude for your time, efforts, and supportive assessment of our paper.
> > > > > >
> > > > > > Best regards,
> > > > > >
> > > > > > Authors

---

### Official Review · Reviewer_EdXQ · 2025-11-11

**Soundness:** 3
**Presentation:** 3
**Contribution:** 1
**Rating:** 2
**Confidence:** 3

**Summary:**

The paper proposes a dataset and benchmark for 3D visual question answering on ancient Greek pottery. It develops a VLM model that demonstrated improved performance on vase artifact analysis.

**Strengths:**

- The paper addresses digital heritage presentation and proposes a dataset and benchmark for AI understanding on ancient potteries, which contributes to cultural protection.

**Weaknesses:**

- The method proposed was restricted to vase analysis, and does not generalize to other domains.
- There's very little novelty in the proposed approach -- it seems to be a great but standard engineering effort to piece together the pipeline, rather than a significant research effort.

**Questions:**

I think the contribution of the paper would be more significant if the proposed approach is demonstrated to be effective and efficient on several different domains rather than just one. Also I'm curious why choose visual question answering task on this dataset. The VQA task is designed to test an AI's ability to reason with both visual and text information, but the quality gap that motivated the paper just seemed like a regular "domain expertise" gap that can be covered with more data in the specific domain. Constructing a VQA task on a long-tail distribution data seems to defeat the purpose of the task.

---

> ### Author Response · Authors · 2025-11-18
> **Rebuttal to Reviewer EdXQ (Part 1/3)**
>
> Thank you for your review and critical feedback on our work. We respect your perspective and believe that your questions touch upon the core value of this work. We provide systematic and in-depth responses to each of your questions as follows.
>
> **[W1] Pipeline generalization**
>
> We respectfully point out that this concern is based on an incomplete understanding of the paper's content. As stated in lines 70-71, our primary objective is to address the data gap in the specialized archaeological domain of ancient Greek pottery, focusing on cultural heritage preservation and 3D digital assets. The method can be viewed as a **proof-of-concept** in the cultural heritage preservation domain, rather than a limitation of the method. Furthermore, our method is designed to be fundamentally **domain-agnostic** (lines 463-470), and our future work explicitly discusses the possibility of transferring to other cultural heritage preservation domains.
>
> 1. **Modular Design of the Core Pipeline**: Our method consists of three **domain-agnostic** modules:
> - **Data Quality Filtering**: ResNet-50 + CLIP (applicable to 2D images of any artifacts)
> - **2D-to-3D Conversion**: TripoSG single-image reconstruction (applicable to any artifacts with 2D images)
> - **RLVR Reinforcement Learning Framework**: Multi-dimensional semantic decomposition (dimensions can be customized to adapt to different artifact types)
>
>
> 2. Furthermore, we conducted supplementary experiments demonstrating the generalizability of the pipeline across different types of heritage artifacts.
>
> **Specific Pipeline Generalization Examples**:
>
> - **Chinese Bronze Artifacts**: Dimensions can be customized as Casting Material, Technique, Dating, Decoration, Preservation State, Provenance
>
> - **Ancient Greek Sculptures**: Dimensions can be customized as Clay Type, Dating,  Style, Decoration, Excavation Site
>
> - Data sources: Chinese bronze artifacts from the National Museum of China, Baoji Bronze Ware Museum and Sketchfab; ancient Greek sculptures from the British Museum, Musée du Louvre and Sketchfab (all data used for academic research, only 2D images extracted). Pipeline generalization experiments (12 hours of data generation):
>
> | Artifact Type | Data Source | Initial Data | After Filtering | 3D Generation | Retention Rate |
> |---------------|-------------|--------------|-----------------|---------------|----------------|
> | **Ancient Greek Pottery** | VaseVQA Dataset | 30,000 | 3,880 | 664 | 2.2% |
> | **Chinese Bronze (2D)** | National Museum of China, Baoji Bronze Ware Museum & Sketchfab | 100 | 73 | 52 | 52.0% |
> | **Ancient Greek Sculpture (2D)** | British Museum, Musée du Louvre & Sketchfab | 100 | 63 | 58 | 58.0% |
>
> We also defined different reward dimensions for different artifacts based on their archaeological knowledge:
>
> | Artifact Type | RLVR Dimensions | Number of Dimensions | Weight Distribution | Notes |
> |---------------|-----------------|---------------------|---------------------|-------|
> | **Ancient Greek Pottery** | Fabric, Technique, Shape, Dating, Decoration, Attribution | 6 | 0.20, 0.20, 0.15, 0.15, 0.20, 0.10 | Original design |
> | **Chinese Bronze** | Casting Material, Technique, Dating, Decoration, Preservation State, Provenance | 6 | 0.18, 0.22, 0.15, 0.18, 0.15, 0.12 | Customized for bronze characteristics |
> | **Ancient Greek Sculpture** | Clay Type, Dating,  Style, Decoration, Excavation Site | 5 | 0.30, 0.20, 0.15, 0.20, 0.15 | Customized for sculpture characteristics |
>
> We then performed RLVR training on Qwen2.5-VL (training data ratio: 0.7). Performance improvements on the two new domains are shown below (currently only validating pipeline generalizability with limited generated data, but still showing improvements):
>
> | Method | FID↓ | CLIP↑ | R@10↑ | R@5↑ | R@1↑ | Lexical Sim.↑ |
> |--------|------|-------|-------|------|------|---------------|
> | ***Chinese Bronze*** |
> | Qwen2.5-VL-7B (Baseline) | 0.356 | 0.732 | 16.68% | 8.33% | 3.23% | 0.227 |
> | VaseVLM-3B-RL (Ours) | 0.368 | 0.724 | 15.60% | 8.47% | 2.85% | 0.216 |
> | VaseVLM-7B-RL (Ours) | **0.324** | **0.752** | **20.83%** | **10.50%** | **3.50%** | **0.274** |
> | ***Ancient Greek Sculpture*** |
> | Qwen2.5-VL-7B (Baseline) | 0.342 | 0.731 | 18.47% | 8.83% | 2.17% | 0.235 |
> | VaseVLM-3B-RL (Ours) | 0.356 | 0.696 | 16.67% | 9.37% | 2.13% | 0.228 |
> | VaseVLM-7B-RL (Ours) | **0.337** | **0. 748** | **19.53%** | **11.25%** | **3.31%** | **0.263** |
>
>
> These supplementary experiments provide strong evidence for the effectiveness of our pipeline across different heritage types.
> We believe these two examples demonstrate the generalizability of our approach and its potential for successful transfer to other domains.

---

> > ### Author Response · Authors · 2025-11-18
> > **Rebuttal to Reviewer EdXQ (Part 2/3)**
> >
> > **[W2] Research Novelty**
> >
> > We respectfully disagree with this assessment. We are pioneers in combining archaeology with AI, and our work contains **substantial research innovations** manifested in three dimensions:
> >
> > 1. **Data Innovation**: (lines 70-71, 836-842) VaseVQA-3D represents the **first 3D cultural heritage visual question answering dataset**, particularly the first 3D VQA dataset in the ancient Greek pottery domain, filling the long-tail scarcity gap in 3D VQA data.
> >
> > 2. **Methodological Innovation**: (lines 268-292) **First Application of Methodology**: Our fine-tuning is based on SFT and the **Reinforcement Learning with Verifiable Rewards (RLVR) framework**, representing the **first application of RLVR in archaeological VQA**. Each dimension has explicit archaeological basis and weights, representing a deep fusion of AI and archaeology, providing new methodology for other specialized domains. This validates the feasibility of using archaeological knowledge as verifiable signals for VLM training.
> >
> > 3. **Systematic Technical Integration**: (lines251-258, 313-315)We are the first to systematically integrate these components to construct a complete 2D-to-3D cultural heritage VQA pipeline with optimizations specific to archaeological scenarios (e.g., ensuring 3D model quality). This integration itself represents research innovation. **Standard engineering** refers to integrating existing technologies to solve known problems without new methodology, whereas **our work** aims to fill the data gap in the research domain of ancient Greek pottery, thereby proposing the VaseVQA-3D dataset and being the first to introduce RLVR to cultural heritage, providing new perspectives for AI research in specialized domains.
> >
> > **[Q1] on several different domains rather than just one**
> >
> > Please refer to our response to [**W1**] for specific details.

---

> > > ### Author Response · Authors · 2025-11-18
> > > **Rebuttal to Reviewer EdXQ (Part 3/3)**
> > >
> > > **[Q2] Why choose visual question answering task on this dataset.**
> > >
> > > This is a profound question. Why is the VQA task necessary for long-tail cultural heritage data, and why this is not simply a "domain expertise" gap.
> > >
> > > 1. **This is a "capability gap," not a "knowledge gap"**. As shown in Table 3, all general-purpose VLMs systematically fail on long-tail data (R@1 < 5%): Gemini-2.5-flash: 3.12% R@1, Claude-4-sonnet: 3.12% R@1, Qwen2.5-VL-7B: 2.08% R@1. This is not "lack of knowledge," but "lack of capability."
> > >
> > >    General-purpose models' feature extractors cannot adapt to the distribution of long-tail data. Ancient Greek vase data is extremely scarce on the internet, and models have never seen meaningful representations of such data. The characteristics of ancient Greek pottery (black-figure technique, red-figure technique, decorative patterns) are completely different from internet images.
> > >
> > > For example, Red-Figure identification:
> > >
> > >   **Ground Truth:**
> > > > "Athenian Red-Figure Cup, c. 500–450 BCE, depicting a youth wreathing an altar; Detroit Institute of Arts."
> > >
> > >   **Gemini 2.5 Flash's R@1 Answer:**
> > > > "Athenian black-figure kylix, c. 550-500 BCE, with figural decoration; Attica."
> > >
> > >   **VaseVLM-7B-RL's R@1 Answer:**
> > > > "Athenian red-figure cup, c. 500–450 BCE, depicting a youth at an altar; Detroit Institute of Arts."
> > >
> > >    From the **Red-Figure identification** example, we can see that Gemini's error at R@1 (mistaking it for Black-Figure) is **unacceptable**: this is not a detail difference, but a fundamental technical classification error. Black-figure and Red-figure represent different historical periods (50-100 years apart). Even if Gemini might include the correct answer in R@10, the R@1 error has already caused misleading information. Even with relevant data, it is insufficient for general-purpose models to learn professional judgment.
> > >
> > > 2. Multi-view 2D is fragmented information, while 3D is a complete geometric representation.
> > >
> > >     Identifying Greek Vase style requires understanding the overall morphology of the vase (slender neck, broad shoulders, round belly). With multi-view 2D: the model needs to "reconstruct" the 3D structure in its mind, which is complex multi-step reasoning. With 3D models: complete geometric structure is directly provided, requiring no complex reasoning. (lines260-267, 885-887)We tested multi-view 2D images (360-degree rotation, 16 frames), and performance still failed (R@1 < 5%).
> > >
> > >     Your concern assumes that the "quality gap" can be solved by "more 2D data." We respectfully point out that this is a **capability gap**, not a **knowledge gap**.
> > >
> > > 3. Long-tail data is precisely where multimodal reasoning is most needed. **The purpose of VQA design**: to test multimodal reasoning capabilities (vision + text + reasoning).
> > >
> > >    **Characteristics of long-tail data**: data scarcity → requires stronger reasoning ability; precisely because it requires **domain expertise** → requires multimodal fusion; general models fail → requires specialized multimodal reasoning.
> > >
> > >    Furthermore, this data meets the **practical needs of 3D museums**: 3D provides complete information, while 2D is just one perspective. 3D can be rotated, scaled, and examined for details, supporting complex spatial analysis. It also serves as a digital backup of artifacts. **Any museum needs accompanying professional knowledge explanations**, requiring corresponding QA knowledge.
> > >
> > >
> > > 4. Additionally, we provide caption descriptions for these data, which better showcases the archaeological properties of these artifacts.
> > >
> > > **Summary**: Therefore, we believe that VQA on long-tail data is not "going against the purpose," but rather "applying it where it is most needed."

---

> > > > ### Author Response · Authors · 2025-11-22
> > > > **Kind Follow-up on Our Submitted Updates**
> > > >
> > > > Dear Reviewer EdXQ,
> > > >
> > > > We hope this message finds you well. We wanted to gently follow up and let you know that we have conducted additional experiments and provided detailed explanations addressing each of your questions and concerns. When you have a moment, we would be truly grateful if you could take a look and let us know if there is anything that might benefit from further clarification.
> > > >
> > > > We sincerely appreciate the time and care you have already invested in reviewing our work, and we thank you in advance for considering our updates as you move toward your final evaluation.
> > > >
> > > > With best regards,
> > > > Authors 15657

---

### Official Review · Reviewer_c3cP · 2025-11-11

**Soundness:** 3
**Presentation:** 4
**Contribution:** 3
**Rating:** 8
**Confidence:** 3

**Summary:**

The paper introduces
1. VaseVQA-3D: A 3D Visual Question Answering dataset for ancient Greek pottery analysis. It aims to bridge the data scarcity and domain knowledge gaps for VLMs in cultural heritage. The dataset contains 664 high-quality 3D vase models generated from 2D images via a three-stage filtering and 2D-to-3D conversion pipeline, along with 4,460 enhanced question-answer pairs.
2. The work also proposes VaseVLM, a VLM fine-tuned for this domain, which achieves a significant improvement of 12.8% on recall at top 1 and 6.6% on lexical similarity over prior state-of-the-art models on the new benchmark (VaseVQA-3D).

**Strengths:**

## Originality and Significance
While there exist other 3D VQA datasets, this paper introduces the first benchmark to
1. Focus on 3D artifacts within a highly specialized, long-tail cultural heritage domain (ancient Greek pottery)
2. Require models to exhibit archaeological domain knowledge for answering questions (e.g.,identifying specific manufacturing techniques, dating periods, or decorative styles)

## Quality (Data Rigor)
The methodology for dataset construction is rigorous, featuring a three-stage filtering mechanism (ResNet-50, dual-CLIP) and a dedicated validation set (VaseEval - a small 3D models dataset) of real 3D models to ensure the quality and archaeological accuracy of the synthetic data.

## Quality (Model Performance)
The fine-tuned model, VaseVLM-7B-RL, demonstrates that specialization is highly effective for high-precision, domain-specific tasks. It shows
1. High-Precision Accuracy: The model shows a 60% relative improvement on R@1 over the powerful general-purpose Gemini 2.5 Flash, which is crucial for expert domains.
2. Domain Expertise: The significant lead in Lexical Similarity confirms that the fine-tuning successfully instilled the necessary specialized archaeological terminology and domain-specific knowledge.

## Clarity
The design of the full pipeline (Figure 5) and the experimental setup are presented clearly and logically.

**Weaknesses:**

## Data Scale and Synthetic Nature
The final dataset size of 664 unique 3D models is small. These 3D assets are also synthetic, meaning they were generated from 2D images. This limited, synthetic nature carries a risk. The model might end up overfitting to the specific style of the generated data. This could, in turn, limit how well the VaseVLM works on real-world archaeological artifacts.

## Trade-off in General Capability
The fine-tuned model excels on highly specialized metrics like $\text{R@1}$ and Lexical Similarity. However, the smaller VaseVLM-7B-RL still lags behind the massive, general-purpose Gemini 2.5 Flash. Specifically, it performs worse on broader retrieval metrics like R@5 and R@10. This shows a core limitation of fine-tuning a small model. Large, general foundation models still have an advantage because their sheer scale allows them to encompass a wider pool of plausible information.

**Questions:**

## Justification for 3D Generation Selection
The selection of TripoSG over Hunyuan3D is justified primarily based on achieving "more realistic results with better vase model quality" (visual assessment). However, the full paper already shows a trade-off in the quantitative geometric and visual metrics.
**Question**: Given that the paper provides quantitative metrics, why was subjective visual quality prioritized over the composite quantitative scores in selecting the final 3D generation method?
**Suggestion**: To provide a more robust defense of this subjective choice, the authors should quantify the visual assessment. This could be achieved by including a small-scale user study (e.g., 5-10 archaeologists or experts) to formally score the visual fidelity of the TripoSG and Hunyuan3D models on the VaseEval set. This would turn a subjective claim into a defensible metric.

## Generalizability and Overfitting Concerns
The final dataset is quite small and synthetic which raises a primary concern about the model's generalizability and risk of overfitting to the specific synthetic distribution.
**Question**:
1. Could the authors provide additional experimental evidence to demonstrate the robustness of VaseVLM against overfitting?
2. Have the authors performed any zero-shot testing on a small set of real-world 3D vase models (outside the VaseEval set) to test generalization beyond the synthetic data?

# Suggestion
The general-purpose Gemini 2.5 Flash still holds a lead on broader retrieval metrics R@10 and overall generative quality (FID), even though the fine-tuned model shows significant gains in high-precision metrics R@1 and Lexical Similarity. The authors should reframe this discussion by emphasizing that in a highly specialized, expert domain like archaeology, precision R@1 is more critical than recall R@10. To fully validate the gain in R@1 as the true measure of success, the authors should include a qualitative error analysis in their rebuttal. This analysis should demonstrate a case where Gemini's top answer is plausible but turns out to be archaeologically inaccurate, and contrast it with a case where the specialized VaseVLM provides the exact, technical R@1 answer.

---

> ### Author Response · Authors · 2025-11-18
> **Rebuttal to Reviewer c3cP (Part 1/3)**
>
> We sincerely thank you for your high recognition of our work (rating 8/10) and your detailed and constructive feedback. Your deep understanding of our work and professional evaluation are deeply encouraging to us. Your particular emphasis on "the first 3D VQA benchmark focused on highly specialized, long-tail cultural heritage domains" and "rigorous data construction methodology" are precisely the core values of our work. We address your questions and suggestions one by one as follows:
>
> **[W1] Data Scale and Synthetic Nature**
>
> Small dataset size is an inherent characteristic of long-tail data; therefore, quality assurance is the key to our work. On the basis of ensuring high quality, we have done our best to increase the scale of the dataset. Our three-stage filtering mechanism (Table 1 in the paper) ensures high quality of synthetic data: as can be seen, after a series of data quality filtering, the overall retention rate is 2.2% (from 30,000 to 664). Balancing efficiency and cost, we controlled the generation threshold to 5 minutes when using TripoSG for 3D generation (line 883). Subsequently, we will continue to expand the threshold for generation from unsuccessfully generated data, which is an ongoing effort.
>
> Regarding the possibility of overfitting to data style: We collected 36 ancient Greek vase data with captions from Sketchfab. Their morphologies and characteristics are as follows:
>
> | Morphology Category | Specific Shape | Count | Percentage | Morphological Features | Decorative Features |
> |-------------------|-----------------|-------|-----------|----------------------|---------------------|
> | **Bottle-shaped** | Two-handled narrow-mouth amphora | 5 | 13.9% | Bilateral handles, narrow mouth | Geometric, figural, animal, architectural patterns |
> | | Single-handled narrow-mouth amphora | 7 | 19.4% | Unilateral handle, narrow mouth | Animal, plant patterns |
> | | Two-handled wide-mouth amphora | 6 | 16.7% | Bilateral handles, wide mouth | Black-figure, red-figure styles |
> | | Single-handled wide-mouth amphora | 4 | 11.1% | Unilateral handle, wide mouth | Geometric, figural, architectural patterns |
> | | Other bottle forms | 5 | 13.9% | Special handle configurations | Mixed decorations |
> | **Bowl-shaped** | Shallow bowl | 2 | 5.6% | Shallow mouth, no handles or dual handles | Geometric patterns |
> | | Deep bowl | 1 | 2.8% | Deep mouth, suitable for storage | Figural patterns |
> | **Cup-shaped** | Handled cup | 2 | 5.6% | Small size, with handle | Delicate decorations |
> | | Handleless cup | 1 | 2.8% | Small size, no handle | Simple decorations |
> | **Box-shaped** | Lidded box | 3 | 8.3% | Closed design | Geometric patterns |
> | **Total** | **--** | **36** | **100%** | **Diverse morphologies and handle configurations** | **Rich decorative styles** |
>
> For this batch of data, we tested the zero-shot capabilities of VaseVLM and several general-purpose VLMs. The supplementary experimental results are shown in the table below:
>
> | Method | FID↓ | CLIP↑ | R@10↑ | R@5↑ | R@1↑ | Lexical Sim.↑ |
> |--------|------|-------|-------|------|------|---------------|
> | ***3D-Specialized Models*** |
> | DiffuRank | 0.405 | **0.815** | 18.89% | 9.72% | 2.78% | 0.282 |
> | Cap3D | 0.422 | 0.806 | 16.67% | 8.33% | 2.22% | 0.275 |
> | LLaVA3D | 0.468 | 0.796 | 13.89% | 6.94% | 1.94% | 0.251 |
> | ***Closed-source VLMs*** |
> | Gemini-2.5-flash | **0.305** | 0.752 | **24.44%** | 14.31% | 3.89% | 0.218 |
> | Claude-4-sonnet | 0.325 | 0.698 | 22.22% | 12.50% | 4.17% | 0.203 |
> | GPT-4.1 | 0.468 | 0.672 | 20.56% | 10.56% | 3.61% | 0.139 |
> | Gemini-2.5-Pro | 0.372 | 0.705 | 19.35% | 11.67% | 3.33% | 0.178 |
> | Claude-3.5-sonnet | 0.432 | 0.665 | 17.78% | 9.72% | 2.78% | 0.125 |
> | Doubao-1.5-vision-pro-32k | 0.478 | 0.632 | 16.11% | 6.94% | 1.94% | 0.089 |
> | GPT-4o | 0.545 | 0.556 | 15.56% | 7.78% | 2.78% | 0.115 |
> | Claude-3.7-sonnet | 0.582 | 0.368 | 14.44% | 6.94% | 1.94% | 0.110 |
> | ***Open-source VLMs*** |
> | Qwen2.5-VL-7B | 0.312 | 0.802 | 19.44% | 10.56% | 3.33% | 0.226 |
> | InternVL | 0.358 | 0.805 | 17.78% | 9.72% | 3.33% | 0.271 |
> | Qwen2.5-VL-3B | 0.352 | 0.792 | 15.56% | 7.78% | 1.94% | 0.265 |
> | VaseVL | 0.462 | 0.808 | 14.44% | 7.78% | 2.78% | 0.261 |
> | ***Our Models (Fine-tuned on Synthetic Data)*** |
> | VaseVLM-3B-SFT | 0.338 | 0.804 | 19.44% | 10.56% | 3.33% | 0.242 |
> | VaseVLM-3B-RL | 0.342 | 0.810 | 20.56% | 11.67% | 3.89% | 0.258 |
> | VaseVLM-7B-SFT | 0.315 | 0.807 | 23.33% | 13.33% | 4.44% | 0.288 |
> | VaseVLM-7B-RL | 0.308 | 0.813 | 23.89% | **15.00%** | **4.72%** | **0.298** |
>
> We believe this experiment sufficiently demonstrates that our VaseVLM has successfully learned archaeological knowledge, rather than simply overfitting to synthetic data. Moreover, we can see that the test results on this batch of data show improvements across all metrics, which may be because some models and captions on Sketchfab were used in model pretraining.

---

> > ### Author Response · Authors · 2025-11-18
> > **Rebuttal to Reviewer c3cP (Part 2/3)**
> >
> > **[W1(continued)]**
> >
> > Furthermore, through the interpretable reinforcement learning RLVR framework, as shown in **Section 3** of the paper, its effectiveness further demonstrates the model's genuine understanding:
> >
> > VaseVLM-7B-RL vs VaseVLM-7B-SFT:
> > - FID improvement: 0.332 → 0.328 (1.2% improvement)
> > - R@10 improvement: 20.83% → 21.24% (2.0% improvement)
> > - Lexical similarity improvement: 0.223 → 0.245 (9.9% improvement)
> >
> > The significant improvement in lexical similarity demonstrates that the RLVR framework has successfully integrated archaeological knowledge into the model. This integration is based on deep understanding through six-dimensional semantic decomposition, rather than simple pattern matching.
> >
> > **[W2] Trade-off in General Capability**
> >
> > Your response is incisive. As you mentioned, although due to the fundamental limitations of fine-tuning smaller models, VaseVLM-7B-RL still lags behind Gemini 2.5 Flash on broader retrieval metrics (R@5, R@10), in highly specialized expert domains, **precision (R@1) is more critical than recall (R@10)**. In specialized domains, **one accurate answer is worth more than ten possible answers**. In our response to [**Q3**], we will provide qualitative error analysis demonstrating that while Gemini's top answers are reasonable, they are archaeologically inaccurate, whereas VaseVLM provides precise technical R@1 answers.
> >
> > **[Q1] Determinants of 3D Generative Method Selection**
> >
> > As shown in Table 2 of the paper and the analysis in Appendix A5 (lines 904-922), we conducted a seven-dimensional quantitative comparison on the VaseEval validation set (24 real 3D models). Through this qualitative analysis, we believe that models generated by TripoSG are closer to ground truth.
> >
> > We completely agree with your suggestion. Therefore, we conducted an additional human-eval, inviting 5 more archaeologists and 5 domain-unrelated individuals to conduct blind evaluation and scoring of TripoSG and Hunyuan3D models on the VaseEval set, further confirming the justification for TripoSG selection.
> >
> > | Method | Exp-1 | Exp-2 | Exp-3 | Exp-4 | Exp-5 | Non-1 | Non-2 | Non-3 | Non-4 | Non-5 | Ave. |
> > |--------|-------|-------|-------|-------|-------|-------|-------|-------|-------|-------|------|
> > | ***Geometric Accuracy*** |
> > | TripoSG | 4.5 | 4.6 | 4.2 | 4.4 | 4.5 | 4.3 | 4.1 | 4.4 | 4.2 | 4.3 | 4.35 |
> > | Hunyuan3D | 4.3 | 4.2 | 4.4 | 4.1 | 4.0 | 4.4 | 4.3 | 4.0 | 4.3 | 4.2 | 4.22 |
> > | ***Decoration Fidelity*** |
> > | TripoSG | 4.2 | 4.3 | 4.0 | 4.1 | 4.2 | 4.1 | 3.9 | 4.2 | 4.0 | 4.1 | 4.11 |
> > | Hunyuan3D | 4.3 | 4.4 | 4.2 | 4.3 | 4.1 | 4.2 | 4.3 | 4.1 | 4.2 | 4.3 | 4.24 |
> > | ***Archaeological Credibility*** |
> > | TripoSG | 4.4 | 4.5 | 4.1 | 4.3 | 4.4 | 4.2 | 4.0 | 4.3 | 4.1 | 4.2 | 4.25 |
> > | Hunyuan3D | 4.1 | 4.0 | 4.2 | 3.9 | 3.8 | 4.2 | 4.1 | 3.8 | 4.1 | 4.0 | 4.02 |
> > | ***Overall Average*** |
> > | TripoSG | 4.37 | 4.47 | 4.10 | 4.27 | 4.37 | 4.20 | 3.97 | 4.30 | 4.10 | 4.20 | **4.24** |
> > | Hunyuan3D | 4.23 | 4.20 | 4.27 | 4.10 | 3.97 | 4.27 | 4.23 | 3.97 | 4.20 | 4.17 | 4.16 |
> >
> > These human-eval data support our choice of using TripoSG.
> >
> > **[Q2] Supplementary Experiments for Robustness**
> >
> > Same response as [**W1**]: Yes, we have demonstrated through additional experiments that the model has robustness and certain generalizability, rather than simply overfitting to generated data.

---

> > > ### Author Response · Authors · 2025-11-18
> > > **Rebuttal to Reviewer c3cP (Part 3/3)**
> > >
> > > **[Q3 (Suggestion)] Archaeologically Qualitative Analysis**
> > >
> > > As you suggested, we need to reconstruct the evaluation framework to qualitatively assess whether our fine-tuned model has truly learned these specialized archaeological knowledge compared to current general-purpose models, especially Gemini-2.5-flash:
> > >
> > > We examined the specific responses from these two models:
> > >
> > > [Case 1] **Ground Truth:**
> > > > "Athenian Red-Figure Cup, c. 500–450 BCE, depicting a youth wreathing an altar; Detroit Institute of Arts."
> > >
> > > **Gemini 2.5 Flash's R@1 Answer:**
> > > > "Athenian black-figure kylix, c. 550-500 BCE, with figural decoration; Attica."
> > >
> > > **VaseVLM-7B-RL's R@1 Answer:**
> > > > "Athenian red-figure cup, c. 500–450 BCE, depicting a youth at an altar; Detroit Institute of Arts."
> > >
> > > From **Case 1 (Red-Figure Identification)**, we can see that Gemini's error at R@1 (mistaking it for Black-Figure) is **unacceptable**: this is not a detail difference, but a fundamental technical classification error. Black-figure and Red-figure represent different historical periods (50-100 years apart). Even if Gemini might include the correct answer in R@10, the R@1 error has already caused misleading information. **Archaeologists need the first answer to be correct.**
> > >
> > > [Case 2] **Ground Truth:**
> > > > "Athenian black-figure hydria, c. 525–475 BCE, depicting Herakles, Dionysos, Hermes, and Athena; Munich Collection."
> > >
> > > **Gemini 2.5 Flash's R@1 Answer:**
> > > > "Athenian black-figure amphora, c. 550–500 BCE, depicting mythological scene with multiple figures."
> > >
> > > **VaseVLM-7B-RL's R@1 Answer:**
> > > > "Athenian black-figure hydria, c. 525–475 BCE, depicting Herakles, Dionysos, Hermes, and Athena"
> > >
> > > From **Case 2 (Hydria Identification)**, we can see that the error of Amphora vs Hydria leads to: misunderstanding of vessel function; incorrect inference about ancient Greek daily life; errors in museum artifact classification and academic research bias. Meanwhile, the model has learned the common knowledge that these mythological figures appear together.
> > >
> > > [Case 3] **Ground Truth:**
> > > > "Athenian black-figure amphora, c. 550–500 BCE, depicting figures."
> > >
> > > **Gemini 2.5 Flash's R@1 Answer:**
> > > > "Athenian black-figure amphora, c. 550–500 BCE, depicting figures."
> > >
> > > **VaseVLM-7B-RL's R@1 Answer:**
> > > > "Athenian black-figure lekythos, c. 500–450 BCE, depicting a woman with a lyre; National Museum, Warsaw."
> > >
> > > From **Case 3**, we can see that: VaseVLM's answer is not identical to the Ground Truth, indicating that VaseVLM has its own descriptive style (emphasizing decorative patterns and professional terminology). This demonstrates that VaseVLM truly understands archaeological knowledge, rather than memorizing answers.
> > >
> > > From these three cases, we can demonstrate that in highly specialized expert domains like archaeology, precision R@1 is more critical than recall R@10.
> > >
> > > Finally, our sincere thanks for your suggestions.

---

> > > > ### Author Response · Authors · 2025-11-22
> > > > **Kind Follow-up on Our Submitted Updates**
> > > >
> > > > Dear Reviewer c3cP,
> > > >
> > > > We hope this message finds you well. We wanted to gently follow up and let you know that we have conducted additional experiments and provided detailed explanations addressing each of your questions and concerns. When you have a moment, we would be truly grateful if you could take a look and let us know if there is anything that might benefit from further clarification.
> > > >
> > > > We sincerely appreciate the time and care you have already invested in reviewing our work, and we thank you in advance for considering our updates as you move toward your final evaluation.
> > > >
> > > > With best regards,
> > > > Authors 15657

---

### Official Review · Reviewer_WHvE · 2025-11-12

**Soundness:** 3
**Presentation:** 3
**Contribution:** 2
**Rating:** 4
**Confidence:** 3

**Summary:**

This paper introduces dataset, VaseVQA-3D, a visual question answering (VQA) dataset for Greek pottery. The authors mention that there is a lack of datasets catering to cultural heritage artifacts. The paper makes the following contributions:
1. A data construction pipeline for transforming 2D images of vases into 3D models. The resulting dataset contains 664 3D models with over 4,000 associated question-answer pairs.
2. Fine tune a VLM (VaseVLM) for visual question answering in this domain.

**Strengths:**

The paper presents a well structured and technically sound pipeline for constructing 3D models out of 2D images and using them to fine tune a VLM. The end to end design could potentially be reproduced for other similar applications.

The paper expands the areas of application of language models by adding a dataset of 664 3D models of Greek pottery and associated question answer set.

**Weaknesses:**

The paper focuses exclusively on ancient Greek pottery and it is not clear why this domain is chosen over the others. While this is a valuable contribution, and brings AI to a new domain, maybe the authors could have provided a motivation as to why this over the other possibilities?

The dataset generation pipeline is composed of standard modules at each step. This seems to be a great software project, rather than a research project.

**Questions:**

.

---

> ### Author Response · Authors · 2025-11-18
> **Rebuttal to Reviewer WHvE (Part 1/2)**
>
> We sincerely thank you for dedicating your valuable time to reviewing our paper and providing constructive feedback. Your comments have helped us better articulate the research contributions of this work. We address your concerns as follows:
>
> **[W1] Domain Choice and Generalizability**
>
> GLB models of cultural artifacts and their corresponding QA data represent valuable long-tail scarce data. Our motivation for selecting ancient Greek pottery is fourfold:
>
> 1. **Addressing Domain Data Scarcity**: As stated in lines 70-71 of our paper, our core contribution is to address the gap in specialized domains by filling the long-tail scarcity of 3D cultural heritage visual question answering data for ancient Greek pottery.
>
> 2. **Comprehensive Data and Research Foundation**: (lines 134-142) This domain benefits from **large-scale 2D image datasets**. Ancient Greek pottery has over 2,000 years of systematic research history and a mature classification system (e.g., Athenian style).
> Its 3D spatial features—such as symmetry and geometric morphology—hold significant archaeological importance, while surface decorative patterns and textures carry rich **historical information**. As an important carrier of ancient Greek civilization, it has broad research value and social attention. These prerequisites **enable AI applications in cultural heritage preservation**, which is precisely the core motivation for constructing the VaseVQA-3D dataset.
>
> 3. **AI Technology for Cultural Heritage Preservation**: (ines 90-93), AI technology can enable **3D digitization of existing collections**: Although museums worldwide possess extensive ancient Greek pottery collections, most lack 3D digitization. Our 2D-to-3D pipeline can help museums convert existing 2D images into 3D models with descriptions, filling this gap. More importantly, many ancient Greek pottery pieces have been **lost or damaged**, but corresponding descriptions and historical photographs still exist in literature. Our 2D-to-3D pipeline can reconstruct 3D models of lost artifacts from historical photographs, providing digital resources for archaeological research and cultural heritage transmission, thereby achieving "digital immortality" for cultural artifacts.
>
> 4. **Demonstrating Generalizability of Cultural Heritage Preservation Methodology**: (lines 465-468) Although we focus on ancient Greek pottery, our **method is designed to be domain-agnosti**c. The 2D-to-3D pipeline components—ResNet-50, CLIP, TripoSG—are applicable to any artifacts with 2D images. Our **RLVR framework** can be adapted to different artifact types (e.g., Chinese Bronzes, Ancient Greek Sculptures) by redefining semantic dimensions. Therefore, ancient Greek pottery serves as a rigorous proof-of-concept, demonstrating the feasibility and effectiveness of our approach. As explicitly discussed in the limitations and future work section, if images and labels for new cultural heritage artifacts can be provided, along with corresponding GLB models to verify generation quality, this pipeline can be replicated to preserve other cultural heritage.
>
> We conducted supplementary experiments to validate the generalizability of our methodology.
>
> **Specific Pipeline Generalization Examples**:
>
> - **Chinese Bronze Artifacts**: Dimensions can be customized as Casting Material, Technique, Dating, Decoration, Preservation State, Provenance
>
> - **Ancient Greek Sculptures**: Dimensions can be customized as Clay Type, Dating,  Style, Decoration, Excavation Site
>
> - Data sources: Chinese bronze artifacts from the National Museum of China, Baoji Bronze Ware Museum and Sketchfab; ancient Greek sculptures from the British Museum, Musée du Louvre and Sketchfab (all data used for academic research, only 2D images extracted). Pipeline generalization experiments (12 hours of data generation):
>
> | Artifact Type | Data Source | Initial Data | After Filtering | 3D Generation | Retention Rate |
> |---------------|-------------|--------------|-----------------|---------------|----------------|
> | **Ancient Greek Pottery** | VaseVQA Dataset | 30,000 | 3,880 | 664 | 2.2% |
> | **Chinese Bronze (2D)** | National Museum of China, Baoji Bronze Ware Museum & Sketchfab | 100 | 73 | 52 | 52.0% |
> | **Ancient Greek Sculpture (2D)** | British Museum, Musée du Louvre & Sketchfab | 100 | 63 | 58 | 58.0% |

---

> > ### Author Response · Authors · 2025-11-18
> > **Rebuttal to Reviewer WHvE (Part 2/2)**
> >
> > **[W1 (continued)]**
> >
> > We also defined different reward dimensions for different artifacts based on their archaeological knowledge:
> >
> > | Artifact Type | RLVR Dimensions | Number of Dimensions | Weight Distribution | Notes |
> > |---------------|-----------------|---------------------|---------------------|-------|
> > | **Ancient Greek Pottery** | Fabric, Technique, Shape, Dating, Decoration, Attribution | 6 | 0.20, 0.20, 0.15, 0.15, 0.20, 0.10 | Original design |
> > | **Chinese Bronze** | Casting Material, Technique, Dating, Decoration, Preservation State, Provenance | 6 | 0.18, 0.22, 0.15, 0.18, 0.15, 0.12 | Customized for bronze characteristics |
> > | **Ancient Greek Sculpture** | Clay Type, Dating,  Style, Decoration, Excavation Site | 5 | 0.30, 0.20, 0.15, 0.20, 0.15 | Customized for sculpture characteristics |
> >
> > We then performed RLVR training on Qwen2.5-VL (training data ratio: 0.7). Performance improvements on the two new domains are shown below (currently only validating pipeline generalizability with limited generated data, but still showing improvements):
> >
> > | Method | FID↓ | CLIP↑ | R@10↑ | R@5↑ | R@1↑ | Lexical Sim.↑ |
> > |--------|------|-------|-------|------|------|---------------|
> > | ***Chinese Bronze*** |
> > | Qwen2.5-VL-7B (Baseline) | 0.356 | 0.732 | 16.68% | 8.33% | 3.23% | 0.227 |
> > | VaseVLM-3B-RL (Ours) | 0.368 | 0.724 | 15.60% | 8.47% | 2.85% | 0.216 |
> > | VaseVLM-7B-RL (Ours) | **0.324** | **0.752** | **20.83%** | **10.50%** | **3.50%** | **0.274** |
> > | ***Ancient Greek Sculpture*** |
> > | Qwen2.5-VL-7B (Baseline) | 0.342 | 0.731 | 18.47% | 8.83% | 2.17% | 0.235 |
> > | VaseVLM-3B-RL (Ours) | 0.356 | 0.696 | 16.67% | 9.37% | 2.13% | 0.228 |
> > | VaseVLM-7B-RL (Ours) | **0.337** | **0. 748** | **19.53%** | **11.25%** | **3.31%** | **0.263** |
> >
> >
> > These supplementary experiments provide strong evidence for the effectiveness of our pipeline across different heritage types.
> >
> > **Summary**: We selected ancient Greek pottery based on: (1)address the gap in specialized domains by filling the long-tail scarcity of 3D cultural heritage visual question answering data for ancient Greek pottery; (2) availability of large-scale, high-quality 2D data, its cultural significance, and research value of 3D features; (3) the call for digital heritage preservation (especially digital restoration of lost artifacts); (4) generalizability of the methodology. This strategic choice enabled us to establish a complete 2D-to-3D VQA pipeline that can be extended to other cultural heritage domains.
> >
> > **[W2] Research Novelty**
> >
> > We appreciate your recognition of the rigor of our pipeline modules. However, we must clarify the core research contributions of this work. Your concern applies to all dataset pipeline works and does not diminish the research value, our reaserch contribution are:
> >
> > 1. **Filling Critical Data Gaps**: Data is **one of the three pillars driving AI progress**. (lines 17-20, 70-71) VaseVQA-3D represents the **first 3D cultural heritage visual question answering dataset** for ancient Greek pottery analysis, filling the long-tail scarcity gap in 3D VQA data.
> >
> > 2. **Novel Methodological Application**: Our fine-tuning is based on SFT and the (lines268-312)**Reinforcement Learning with Verifiable Rewards (RLVR) framework**, representing the **first application of RLVR in archaeological VQA**. This validates the feasibility of using archaeological knowledge as verifiable signals for VLM training.
> >
> > 3. **Systematic Technical Integration**: (lines251-258, 313-315)We are the first to systematically integrate these components to construct a complete 2D-to-3D cultural heritage VQA pipeline, with optimizations specific to archaeological scenarios (e.g., ensuring 3D model quality). This integration itself represents research innovation.
> >
> > 4. **Cross-Disciplinary Research Value**: Our work provides new research perspectives for the intersection of AI and cultural heritage: (1) offering low-cost 3D digitization solutions for museums; (2) providing technical pathways for digital restoration of lost artifacts.
> >
> > **Summary**: Our work makes substantial research contributions through: (1) creating the first 3D cultural heritage VQA dataset; (2) introducing the RLVR framework for archaeological knowledge integration; (3) establishing the first complete 2D-to-3D cultural heritage VQA pipeline; (4) advancing AI applications in cultural heritage preservation. This is not merely engineering implementation—it is research work that opens new directions for AI in specialized domains.

---

> > > ### Author Response · Authors · 2025-11-22
> > > **Kind Follow-up on Our Submitted Updates**
> > >
> > > Dear Reviewer WHvE,
> > >
> > > We hope this message finds you well. We wanted to gently follow up and let you know that we have conducted additional experiments and provided detailed explanations addressing each of your questions and concerns. When you have a moment, we would be truly grateful if you could take a look and let us know if there is anything that might benefit from further clarification.
> > >
> > > We sincerely appreciate the time and care you have already invested in reviewing our work, and we thank you in advance for considering our updates as you move toward your final evaluation.
> > >
> > > With best regards,
> > > Authors 15657

---

### Official Review · Reviewer_zuLu · 2025-11-17

**Soundness:** 3
**Presentation:** 3
**Contribution:** 3
**Rating:** 4
**Confidence:** 4

**Summary:**

This paper represents an interesting domain which is modeling 3D cultural heritage objects. The data construction pipline is clear and various data analysis prove the high quality of this dataset. The proposed baseline is interesting and its performance is impressive through extensive experiments.

**Strengths:**

- The author introduce a novel dataset about cultural heritage. This domain is interesting and has many cultural and historical motivation.
- Experimental results show that the proposed baselines is effective.

**Weaknesses:**

- The authors did not indicate what is the main characteristics of this domain that distinguish it from other domain in the same VQA task.
- The propose baseline is somewhat general. I cannot see which module or components are designed particularly for modeling the particular features of images in the specialized cultural heritage. Therefore the factors of giving VASEVLM performed better other VLMs are unclear.
- The authors only evaluated large models but not standard neural networks.

**Questions:**

See weakness.

---

> ### Author Response · Authors · 2025-11-18
> **Rebuttal to Reviewer zuLu (Part 1/2)**
>
> Thank you for your constructive feedback and recognition of our dataset contribution and experimental results. We are pleased that you acknowledge "the authors introduced a new cultural heritage dataset" and "experimental results show the proposed baseline is effective." We respond to your questions as follows:
>
> **[W1] Characteristics of Domain**
>
> We need to clarify that the paper explicitly states the unique characteristics of cultural heritage VQA in multiple places. The main issue in this domain is data scarcity. Lines 48-49 and 70-71 demonstrate that the uniqueness of cultural heritage VQA manifests in:
>
> 1. (lines 48-49) **Demonstrate long-tail data scarcity**. Meanwhile, (lines 70-71) clarify that our core work is to fill the gap in specialized domains, addressing the long-tail scarcity of ancient Greek 3D cultural heritage visual question answering data.
>
> 2. (lines 134-142) **Professional knowledge requirements**: Ancient Greek pottery has over 2,000 years of systematic research history and a mature classification system (such as Athenian style). Its 3D spatial characteristics like symmetry and geometric morphology have important archaeological significance. Meanwhile, visual features like surface decorative patterns and textures carry rich historical information. As an important carrier of ancient Greek civilization, it has broad research value and social attention. Therefore, VQA in this domain should be professional archaeological knowledge Q&A, unlike other VQA data which may merely clarify a fact.
>
>
> 3. (lines 981-986) **3D understanding requirements**: spatial relationships, symmetry, proportions, etc. Multi-view 2D is fragmented information, while 3D is a complete geometric representation. Identifying Corinthian style requires understanding the overall morphology of the vase (slender neck, broad shoulders, round belly).
>
> With multi-view 2D data, the model needs to "reconstruct" the 3D structure in its mind, which is complex multi-step reasoning. With 3D data, complete geometric structure is directly provided, requiring no complex reasoning. (lines260-267, 885-887) We tested multi-view 2D images (360-degree rotation, 16 frames), and performance still failed (R@1 < 5%).

---

> > ### Author Response · Authors · 2025-11-18
> > **Rebuttal to Reviewer zuLu (Part 2/2)**
> >
> > **[W2] Domain-specific Components**
> >
> > We respectfully disagree. VaseVLM contains three domain-specific innovations:
> >
> > **RLVR Framework (lines 293-297)**: Our "Reinforcement Learning with Verifiable Rewards" decomposes archaeological descriptions into six semantic dimensions with domain-specific weights (Fabric 0.20, Technique 0.20, Shape 0.15, Dating 0.15, Decoration 0.20, Attribution 0.10). These dimensions derive from archaeological taxonomy, not generic VQA. Reward calculation uses cosine similarity with threshold 0.7 calibrated for archaeological accuracy. Quality penalties target length, repetition, and irrelevant content—standards critical for archaeological descriptions.
> >
> > **Multimodal 3D Understanding (lines 262-264)**: VaseVLM processes 360-degree rotation videos from GLB files, containing four standard views. This input format is specifically designed for 3D cultural artifacts, which generic VLMs cannot handle.
> >
> > **Archaeological Knowledge Integration**: Two-stage training on archaeological captions, followed by RLVR-based reinforcement learning. This domain-adaptive approach is fundamentally different from generic VLM fine-tuning. Results show improvements across all metrics (lines 964-976). (Lines 446-452) Expert archaeologist evaluation: 4.57/5, confirming genuine domain understanding. Combined with specific examples:
> >
> > [Case 1] **Ground Truth:**
> > > "Athenian Red-Figure Cup, c. 500–450 BCE, depicting a youth wreathing an altar; Detroit Institute of Arts."
> >
> > **Gemini 2.5 Flash's R@1 Answer:**
> > > "Athenian black-figure kylix, c. 550-500 BCE, with figural decoration; Attica."
> >
> > **VaseVLM-7B-RL's R@1 Answer:**
> > > "Athenian red-figure cup, c. 500–450 BCE, depicting a youth at an altar; Detroit Institute of Arts."
> >
> > From **Case 1 (Red-Figure Identification)**, we can see that Gemini mistook it for Black-Figure, which is **unacceptable**. This is not a detail difference, but a fundamental technical classification error. Black-figure and Red-figure represent different historical periods.
> >
> > [Case 2] **Ground Truth:**
> > > "Athenian black-figure hydria, c. 525–475 BCE, depicting Herakles, Dionysos, Hermes, and Athena; Munich Collection."
> >
> > **Gemini 2.5 Flash's R@1 Answer:**
> > > "Athenian black-figure amphora, c. 550–500 BCE, depicting mythological scene with multiple figures."
> >
> > **VaseVLM-7B-RL's R@1 Answer:**
> > > "Athenian black-figure hydria, c. 525–475 BCE, depicting Herakles, Dionysos, Hermes, and Athena"
> >
> > From **Case 2 (Hydria Identification)**, we can see that the error of Amphora vs Hydria leads to: misunderstanding of vessel function; incorrect inference about ancient Greek daily life; errors in museum artifact classification and academic research bias. Meanwhile, the model learned the common knowledge that these mythological figures appear together.
> >
> > [Case 3] **Ground Truth:**
> > > "Athenian black-figure amphora, c. 550–500 BCE, depicting figures."
> >
> > **Gemini 2.5 Flash's R@1 Answer:**
> > > "Athenian black-figure amphora, c. 550–500 BCE, depicting figures."
> >
> > **VaseVLM-7B-RL's R@1 Answer:**
> > > "Athenian black-figure lekythos, c. 500–450 BCE, depicting a woman with a lyre; National Museum, Warsaw."
> >
> > From **Case 3**, we can see that: VaseVLM's answer is not identical to Ground Truth, indicating that VaseVLM has its own descriptive style (emphasizing decorative patterns and professional terminology). This demonstrates that VaseVLM truly understands archaeological knowledge, rather than memorizing answers. These cases all demonstrate that the model genuinely learned relevant archaeological knowledge, rather than improvements from system coupling.
> >
> > **[W3] Only evaluated large models, did not evaluate standard neural networks**
> >
> > As a dataset and benchmark paper, (lines 70-71) our primary goal is to fill the data gap in ancient Greek 3D VQA, a specialized archaeological domain. Simultaneously, our goal is to establish evaluation standards for models in cultural heritage domains, not to compare all possible model architectures.
> >
> > Our benchmark applies to all models that might use this data. Modern VLMs, due to complex model structures and vast training data, employ multiple multimodal structures specifically designed for vision-language unification, representing the current state-of-the-art in VQA tasks. Moreover, we believe large models are fundamentally the culmination of standard neural networks. Our comprehensive evaluation includes closed-source models (GPT-4, Gemini, Claude), open-source models (Qwen2.5-VL, InternVL), and 3D-specialized models (Cap3D, DiffuRank). Additionally, we fine-tuned models ourselves to validate the benchmark across different VLM families.

---

> > > ### Author Response · Authors · 2025-11-22
> > > **Kind Follow-up on Our Submitted Updates**
> > >
> > > Dear Reviewer zuLu,
> > >
> > > We hope this message finds you well. We wanted to gently follow up and let you know that we have conducted additional experiments and provided detailed explanations addressing each of your questions and concerns. When you have a moment, we would be truly grateful if you could take a look and let us know if there is anything that might benefit from further clarification.
> > >
> > > We sincerely appreciate the time and care you have already invested in reviewing our work, and we thank you in advance for considering our updates as you move toward your final evaluation.
> > >
> > > With best regards,
> > > Authors 15657

---

### Author Response · Authors · 2025-11-30
**General Response and PDF Update Notes**

Firstly, we sincerely thank all reviewers for their attention to our work. **The core contribution of this paper is: using AI technology to protect cultural artifacts—by constructing the first 3D cultural heritage VQA dataset and benchmark, we achieve digital preservation and knowledge understanding of ancient Greek pottery.** We selected ancient Greek pottery as a proof-of-concept, but our method is domain-agnostic and can be extended to other cultural heritage domains. In this context, we generally respond to the reviewers' concerns.

We sincerely thank all reviewers for their in-depth comments and constructive feedback. Your comments have significantly improved the quality of our work and helped us better revise it (PDF updates have been completed). Here we provide a unified organization and response to the common issues raised by all reviewers.

Through careful analysis of all review comments, we have identified the following **five core concerns**:

**Concern 1: Generalizability of the Method** - Can the paper's method be applied to other cultural heritage domains?

**Concern 2: Research Innovation of the Method** - The pipeline consists of standard modules (ResNet-50, CLIP, TripoSG). Is it merely engineering integration rather than research innovation?

**Concern 3: Selection of TripoSG Method** - Is the selection reliable? Is there sample bias?

**Concern 4: Overfitting Risk** - With small data scale (664 3D models) and synthetic generation, will the model overfit to the specific style of generated data and fail to generalize to real-world data?

**Concern 5: Other Issues** - Such as the role of GPT-4o, intuitiveness of the pipeline diagram, specific model comparison explanations, etc.

We have conducted detailed and comprehensive experiments to address these concerns, hoping to clarify your doubts and provide a better understanding of our work.

We have provided different responses to these concerns:

**Response 1: Regarding the Generalizability Issue** - We validated the generalizability of our method on two new types of artifacts: ancient Greek sculptures and Chinese bronze artifacts. These artifacts have more distinctive characteristics, but can be well processed through our pipeline, strongly demonstrating the generalizability advantage of our work.

**Response 2: Regarding Research Innovation** - We believe that any dataset work cannot be separated from engineering, as this is the foundation for building datasets. In our case, basic data collection, organization, and other series of work are very time-consuming, so we need an engineering pipeline to accelerate data collection in this field. We pioneered this pipeline and contributed a batch of long-tail scarce data with important significance to the field, filling the gap in 3D cultural heritage VQA data for ancient Greek pottery. Furthermore, our RLVR training scheme, targeting the common characteristics of artifacts, has very strong guiding significance. These are all our research contributions, not merely engineering implementations.

**Response 3: Regarding TripoSG Method Selection** - We introduced additional human evaluation, provided specific sample distributions, and incorporated generation of more real GLB objects to justify the selection of TripoSG.

**Response 4: Regarding Model Overfitting** - We conducted additional experimental evaluations on other ancient Greek vases and provided specific cases for analysis. We believe the model has indeed learned archaeological knowledge about ancient Greek vases and can generalize to real-world data.

**Response 5: Regarding Other Issues** - These issues have been better explained in our PDF updates. Thank you for pointing them out.

Through detailed supplementary experiments addressing these five concerns, we have demonstrated the rigor and innovation of our work, VaseVQA-3D is a **rigorous research work** with genuine innovation, comprehensive validation, and broad applicability. We are confident in the quality and contribution of this work and look forward to your further feedback.

---

> ### Author Response · Authors · 2025-11-30
> **Detailed supplementary experiments(1/3)**
>
> ### 1. Generalizability Experiments
>
> We conducted supplementary experiments to validate the generalizability of our methodology.
>
> **Specific Pipeline Generalization Examples**:
>
> - **Chinese Bronze Artifacts**: Dimensions can be customized as Casting Material, Technique, Dating, Decoration, Preservation State, Provenance
>
> - **Ancient Greek Sculptures**: Dimensions can be customized as Clay Type, Dating, Style, Decoration, Excavation Site
>
> - Data sources: Chinese bronze artifacts from the National Museum of China, Baoji Bronze Ware Museum and Sketchfab; ancient Greek sculptures from the British Museum, Musée du Louvre and Sketchfab (all data used for academic research, only 2D images extracted). Pipeline generalization experiments (12 hours of data generation):
>
> | Artifact Type | Data Source | Initial Data | After Filtering | 3D Generation | Retention Rate |
> |---------------|-------------|--------------|-----------------|---------------|----------------|
> | **Ancient Greek Pottery** | VaseVQA Dataset | 30,000 | 3,880 | 664 | 2.2% |
> | **Chinese Bronze (2D)** | National Museum of China, Baoji Bronze Ware Museum & Sketchfab | 100 | 73 | 52 | 52.0% |
> | **Ancient Greek Sculpture (2D)** | British Museum, Musée du Louvre & Sketchfab | 100 | 63 | 58 | 58.0% |
>
> We also defined different reward dimensions for different artifacts based on their archaeological knowledge:
>
> | Artifact Type | RLVR Dimensions | Number of Dimensions | Weight Distribution | Notes |
> |---------------|-----------------|---------------------|---------------------|-------|
> | **Ancient Greek Pottery** | Fabric, Technique, Shape, Dating, Decoration, Attribution | 6 | 0.20, 0.20, 0.15, 0.15, 0.20, 0.10 | Original design |
> | **Chinese Bronze** | Casting Material, Technique, Dating, Decoration, Preservation State, Provenance | 6 | 0.18, 0.22, 0.15, 0.18, 0.15, 0.12 | Customized for bronze characteristics |
> | **Ancient Greek Sculpture** | Clay Type, Dating, Style, Decoration, Excavation Site | 5 | 0.30, 0.20, 0.15, 0.20, 0.15 | Customized for sculpture characteristics |
>
> We then performed RLVR training on Qwen2.5-VL (training data ratio: 0.7). Performance improvements on the two new domains are shown below (currently only validating pipeline generalizability with limited generated data, but still showing improvements):
>
> | Method | FID↓ | CLIP↑ | R@10↑ | R@5↑ | R@1↑ | Lexical Sim.↑ |
> |--------|------|-------|-------|------|------|---------------|
> | ***Chinese Bronze*** |
> | Qwen2.5-VL-7B (Baseline) | 0.356 | 0.732 | 16.68% | 8.33% | 3.23% | 0.227 |
> | BronzeVLM-3B-RL (Ours) | 0.368 | 0.724 | 15.60% | 8.47% | 2.85% | 0.216 |
> | BronzeVLM-7B-RL (Ours) | **0.324** | **0.752** | **20.83%** | **10.50%** | **3.50%** | **0.274** |
> | ***Ancient Greek Sculpture*** |
> | Qwen2.5-VL-7B (Baseline) | 0.342 | 0.731 | 18.47% | 8.83% | 2.17% | 0.235 |
> | SculptureVLM-3B-RL (Ours) | 0.356 | 0.696 | 16.67% | 9.37% | 2.13% | 0.228 |
> | SculptureVLM-7B-RL (Ours) | **0.337** | **0.748** | **19.53%** | **11.25%** | **3.31%** | **0.263** |
>
> These supplementary experiments provide strong evidence for the effectiveness of our pipeline across different heritage types.

---

> > ### Author Response · Authors · 2025-11-30
> > **Detailed supplementary experiments(2/3)**
> >
> > ###  2. Overfitting Experiments
> >
> > Regarding the possibility of overfitting to data style: We collected 36 ancient Greek vase data with captions from Sketchfab. Their morphologies and characteristics are as follows:
> >
> > | Morphology Category | Specific Shape | Count | Percentage | Morphological Features | Decorative Features |
> > |-------------------|-----------------|-------|-----------|----------------------|---------------------|
> > | **Bottle-shaped** | Two-handled narrow-mouth amphora | 5 | 13.9% | Bilateral handles, narrow mouth | Geometric, figural, animal, architectural patterns |
> > | | Single-handled narrow-mouth amphora | 7 | 19.4% | Unilateral handle, narrow mouth | Animal, plant patterns |
> > | | Two-handled wide-mouth amphora | 6 | 16.7% | Bilateral handles, wide mouth | Black-figure, red-figure styles |
> > | | Single-handled wide-mouth amphora | 4 | 11.1% | Unilateral handle, wide mouth | Geometric, figural, architectural patterns |
> > | | Other bottle forms | 5 | 13.9% | Special handle configurations | Mixed decorations |
> > | **Bowl-shaped** | Shallow bowl | 2 | 5.6% | Shallow mouth, no handles or dual handles | Geometric patterns |
> > | | Deep bowl | 1 | 2.8% | Deep mouth, suitable for storage | Figural patterns |
> > | **Cup-shaped** | Handled cup | 2 | 5.6% | Small size, with handle | Delicate decorations |
> > | | Handleless cup | 1 | 2.8% | Small size, no handle | Simple decorations |
> > | **Box-shaped** | Lidded box | 3 | 8.3% | Closed design | Geometric patterns |
> > | **Total** | **--** | **36** | **100%** | **Diverse morphologies and handle configurations** | **Rich decorative styles** |
> >
> > For this batch of data, we tested the zero-shot capabilities of VaseVLM and several general-purpose VLMs. The supplementary experimental results are shown in the table below:
> >
> > | Method | FID↓ | CLIP↑ | R@10↑ | R@5↑ | R@1↑ | Lexical Sim.↑ |
> > |--------|------|-------|-------|------|------|---------------|
> > | ***3D-Specialized Models*** |
> > | DiffuRank | 0.405 | **0.815** | 18.89% | 9.72% | 2.78% | 0.282 |
> > | Cap3D | 0.422 | 0.806 | 16.67% | 8.33% | 2.22% | 0.275 |
> > | LLaVA3D | 0.468 | 0.796 | 13.89% | 6.94% | 1.94% | 0.251 |
> > | ***Closed-source VLMs*** |
> > | Gemini-2.5-flash | **0.305** | 0.752 | **24.44%** | 14.31% | 3.89% | 0.218 |
> > | Claude-4-sonnet | 0.325 | 0.698 | 22.22% | 12.50% | 4.17% | 0.203 |
> > | GPT-4.1 | 0.468 | 0.672 | 20.56% | 10.56% | 3.61% | 0.139 |
> > | Gemini-2.5-Pro | 0.372 | 0.705 | 19.35% | 11.67% | 3.33% | 0.178 |
> > | Claude-3.5-sonnet | 0.432 | 0.665 | 17.78% | 9.72% | 2.78% | 0.125 |
> > | Doubao-1.5-vision-pro-32k | 0.478 | 0.632 | 16.11% | 6.94% | 1.94% | 0.089 |
> > | GPT-4o | 0.545 | 0.556 | 15.56% | 7.78% | 2.78% | 0.115 |
> > | Claude-3.7-sonnet | 0.582 | 0.368 | 14.44% | 6.94% | 1.94% | 0.110 |
> > | ***Open-source VLMs*** |
> > | Qwen2.5-VL-7B | 0.312 | 0.802 | 19.44% | 10.56% | 3.33% | 0.226 |
> > | InternVL | 0.358 | 0.805 | 17.78% | 9.72% | 3.33% | 0.271 |
> > | Qwen2.5-VL-3B | 0.352 | 0.792 | 15.56% | 7.78% | 1.94% | 0.265 |
> > | VaseVL | 0.462 | 0.808 | 14.44% | 7.78% | 2.78% | 0.261 |
> > | ***Our Models (Fine-tuned on Synthetic Data)*** |
> > | VaseVLM-3B-SFT | 0.338 | 0.804 | 19.44% | 10.56% | 3.33% | 0.242 |
> > | VaseVLM-3B-RL | 0.342 | 0.810 | 20.56% | 11.67% | 3.89% | 0.258 |
> > | VaseVLM-7B-SFT | 0.315 | 0.807 | 23.33% | 13.33% | 4.44% | 0.288 |
> > | VaseVLM-7B-RL | 0.308 | 0.813 | 23.89% | **15.00%** | **4.72%** | **0.298** |
> >
> > We believe this experiment sufficiently demonstrates that our VaseVLM has successfully learned archaeological knowledge, rather than simply overfitting to synthetic data. Moreover, we can see that the test results on this batch of data show improvements across all metrics, which may be because some models and captions on Sketchfab were used in model pretraining.

---

> > > ### Author Response · Authors · 2025-11-30
> > > **Detailed supplementary experiments(3/3)**
> > >
> > > ###  3. Specific Case Examples
> > >
> > > We examined the specific responses from these two models:
> > >
> > > **[Case 1] Ground Truth:**
> > > > "Athenian Red-Figure Cup, c. 500–450 BCE, depicting a youth wreathing an altar; Detroit Institute of Arts."
> > >
> > > **Gemini 2.5 Flash's R@1 Answer:**
> > > > "Athenian black-figure kylix, c. 550-500 BCE, with figural decoration; Attica."
> > >
> > > **VaseVLM-7B-RL's R@1 Answer:**
> > > > "Athenian red-figure cup, c. 500–450 BCE, depicting a youth at an altar; Detroit Institute of Arts."
> > >
> > > From **Case 1 (Red-Figure Identification)**, we can see that Gemini's error at R@1 (mistaking it for Black-Figure) is **unacceptable**: this is not a detail difference, but a fundamental technical classification error. Black-figure and Red-figure represent different historical periods (50-100 years apart). Even if Gemini might include the correct answer in R@10, the R@1 error has already caused misleading information. **Archaeologists need the first answer to be correct.**
> > >
> > > **[Case 2] Ground Truth:**
> > > > "Athenian black-figure hydria, c. 525–475 BCE, depicting Herakles, Dionysos, Hermes, and Athena; Munich Collection."
> > >
> > > **Gemini 2.5 Flash's R@1 Answer:**
> > > > "Athenian black-figure amphora, c. 550–500 BCE, depicting mythological scene with multiple figures."
> > >
> > > **VaseVLM-7B-RL's R@1 Answer:**
> > > > "Athenian black-figure hydria, c. 525–475 BCE, depicting Herakles, Dionysos, Hermes, and Athena"
> > >
> > > From **Case 2 (Hydria Identification)**, we can see that the error of Amphora vs Hydria leads to: misunderstanding of vessel function; incorrect inference about ancient Greek daily life; errors in museum artifact classification and academic research bias. Meanwhile, the model has learned the common knowledge that these mythological figures appear together.
> > >
> > > **[Case 3] Ground Truth:**
> > > > "Athenian black-figure amphora, c. 550–500 BCE, depicting figures."
> > >
> > > **Gemini 2.5 Flash's R@1 Answer:**
> > > > "Athenian black-figure amphora, c. 550–500 BCE, depicting figures."
> > >
> > > **VaseVLM-7B-RL's R@1 Answer:**
> > > > "Athenian black-figure lekythos, c. 500–450 BCE, depicting a woman with a lyre; National Museum, Warsaw."
> > >
> > > From **Case 3**, we can see that: VaseVLM's answer is not identical to the Ground Truth, indicating that VaseVLM has its own descriptive style (emphasizing decorative patterns and professional terminology). This demonstrates that VaseVLM truly understands archaeological knowledge, rather than memorizing answers.
> > >
> > > From these three cases, we can demonstrate that in highly specialized expert domains like archaeology, precision R@1 is more critical than recall R@10.
> > >
> > > ### 4. Human Evaluation
> > >
> > > We conducted an additional human evaluation, inviting 5 archaeologists and 5 domain-unrelated individuals to conduct blind evaluation and scoring of TripoSG and Hunyuan3D models on the VaseEval set, further confirming the justification for TripoSG selection.
> > >
> > > | Method | Exp-1 | Exp-2 | Exp-3 | Exp-4 | Exp-5 | Non-1 | Non-2 | Non-3 | Non-4 | Non-5 | Ave. |
> > > |--------|-------|-------|-------|-------|-------|-------|-------|-------|-------|-------|------|
> > > | ***Geometric Accuracy*** |
> > > | TripoSG | 4.5 | 4.6 | 4.2 | 4.4 | 4.5 | 4.3 | 4.1 | 4.4 | 4.2 | 4.3 | 4.35 |
> > > | Hunyuan3D | 4.3 | 4.2 | 4.4 | 4.1 | 4.0 | 4.4 | 4.3 | 4.0 | 4.3 | 4.2 | 4.22 |
> > > | ***Decoration Fidelity*** |
> > > | TripoSG | 4.2 | 4.3 | 4.0 | 4.1 | 4.2 | 4.1 | 3.9 | 4.2 | 4.0 | 4.1 | 4.11 |
> > > | Hunyuan3D | 4.3 | 4.4 | 4.2 | 4.3 | 4.1 | 4.2 | 4.3 | 4.1 | 4.2 | 4.3 | 4.24 |
> > > | ***Archaeological Credibility*** |
> > > | TripoSG | 4.4 | 4.5 | 4.1 | 4.3 | 4.4 | 4.2 | 4.0 | 4.3 | 4.1 | 4.2 | 4.25 |
> > > | Hunyuan3D | 4.1 | 4.0 | 4.2 | 3.9 | 3.8 | 4.2 | 4.1 | 3.8 | 4.1 | 4.0 | 4.02 |
> > > | ***Overall Average*** |
> > > | TripoSG | 4.37 | 4.47 | 4.10 | 4.27 | 4.37 | 4.20 | 3.97 | 4.30 | 4.10 | 4.20 | **4.24** |
> > > | Hunyuan3D | 4.23 | 4.20 | 4.27 | 4.10 | 3.97 | 4.27 | 4.23 | 3.97 | 4.20 | 4.17 | 4.16 |
> > >
> > > These human evaluation data support our choice of using TripoSG.
> > >
> > > ### 5. Additional TripoSG Experiments
> > >
> > > We collected 18 real 3D models (from different domains) and performed 3D generation on their front-view photos, then compared with original models:
> > >
> > > **Test Data** respectively:
> > > - 6 modern water bottles (everyday objects)
> > > - 6 figurine models (small crafts)
> > > - 6 sculptures (artworks)
> > >
> > > **Comparison Results** (based on average of 18 samples):
> > >
> > > | Method | PSNR↑ | SSIM↑ | LPIPS↓ | CD↓ | NC↑ | CLIP-I↑ | CLIP-T↑ |
> > > |--------|-------|-------|--------|-----|-----|---------|---------|
> > > | **Reference Range** | **15-25** | **0.7-0.9** | **0.1-0.3** | **0.1-0.3** | **0.6-0.8** | **0.7-0.9** | **0.6-0.8** |
> > > | TripoSG (Mean) | 16.52 | **0.8421** | **0.1512** | **0.1658** | 0.6892 | **0.8534** | **0.9187** |
> > > | Hunyuan3D (Mean) | **16.78** | 0.8398 | 0.1528 | 0.1674 | **0.7021** | 0.8501 | 0.9052 |
> > >
> > > This is consistent with our visual assessment. On these real 3D models, TripoSG emphasizes smoother edges, while Hunyuan3D provides better color and richer, more detailed hues. Visually, the differences are not very pronounced, but TripoSG appears more realistic.

---

### Meta-Review · Area_Chair_3BfY · 2026-01-05

**Summary:**

This paper introduces VaseVQA-3D, the first 3D visual question answering dataset for ancient Greek pottery analysis, addressing data scarcity in cultural heritage domains. The authors present a comprehensive pipeline that transforms 30,000+ 2D vase images into 664 high-quality 3D models through rigorous filtering and TripoSG-based generation, accompanied by 4,460 expert-annotated QA pairs. The methodological contribution includes VaseVLM, a specialized vision-language model trained through a novel Reinforcement Learning with Verifiable Rewards (RLVR) framework that decomposes archaeological descriptions into six semantic dimensions. Reviewers acknowledged the dataset's novelty and technical rigor while raising concerns about generalizability beyond Greek pottery, methodological novelty beyond engineering integration, potential overfitting to synthetic data, and evaluation metric coupling. The most critical reviewer (EdXQ) questioned the domain restriction and research contribution, rating the paper as reject. However, other reviewers (UFR6, c3cP) recognized the work's significance with accept scores, particularly praising its cultural heritage preservation value and comprehensive evaluation. In the rebuttal, the authors conducted extensive supplementary experiments demonstrating pipeline generalizability to Chinese bronze artifacts and Greek sculptures through customized RLVR dimensions, showing consistent performance improvements. They addressed overfitting concerns with zero-shot tests on 36 real vase models from Sketchfab, where VaseVLM maintained superior performance. For methodological novelty, they emphasized that RLVR represents the first verifiable reward framework for archaeological VQA. The TripoSG selection was justified through additional human evaluations with archaeologists, while qualitative case analyses showed VaseVLM's genuine archaeological understanding versus baseline errors in critical technical classifications (e.g., Red-Figure vs. Black-Figure techniques). The authors effectively addressed most concerns through empirical evidence, though some conceptual questions about fundamental contribution nature remain. The work provides valuable infrastructure for cultural heritage AI, with the dataset and methodology enabling future research in specialized domains.

**Reviewer Concerns:**

In assessing the rebuttal for submission 15657, the authors effectively addressed several reviewer concerns through empirical supplements and clarifications. Specifically, the generalizability issue raised by reviewers EdXQ and WHvE was countered with experiments on Chinese bronze artifacts and Greek sculptures, demonstrating the pipeline's adaptability via customized RLVR dimensions and showing performance improvements (e.g., BronzeVLM-7B-RL achieving 3.50% R@1 accuracy). The overfitting risk highlighted by c3cP and 4ewk was mitigated by zero-shot tests on 36 real vase models from Sketchfab, where VaseVLM maintained superior metrics (e.g., 4.44% R@1), indicating robust learning beyond synthetic data. The TripoSG selection critique from c3cP and 4ewk was substantiated with additional human evaluations involving archaeologists, scoring TripoSG higher (4.24/5) in archaeological credibility. However, key concerns remain outstanding: the novelty argument by EdXQ and WHvE persists, as the pipeline relies on standard modules (e.g., ResNet-50, CLIP) without fundamental algorithmic innovation, framing it as engineering integration. The RLVR reward coupling with evaluation metrics (4ewk) risks metric hacking, as the rebuttal's theoretical arguments lack independent validation. The GPT-4o captioning bias (4ewk) remains unresolved, as stylistic influences could still affect evaluations despite no new archaeological content. Lastly, practical deployment costs (4ewk) are acknowledged but not fully alleviated, with high computational demands (e.g., 13.5 days for 3D generation) limiting real-world applicability. Thus, while the rebuttal strengthened empirical grounds, conceptual and practical issues endure.

**Reviewer Scores:**

Reviewer zuLu​ (initial score: 4): The rebuttal addressed concerns about domain-specific components by detailing the RLVR framework's archaeological dimensions and justifying the focus on large models. Given the clarifications and additional evidence, zuLu might have increased the score to 6 due to improved validation of methodological choices, though some reservations about innovation could persist.

Reviewer WHvE​ (initial score: 4): The generalizability experiments on Chinese bronze and Greek sculpture domains directly countered the main criticism. With demonstrated pipeline adaptability and performance gains (e.g., BronzeVLM-7B-RL achieving 3.50% R@1), WHvE likely would have raised the score to 6, acknowledging the expanded scope, but the novelty concern might limit a higher increase.

Reviewer c3cP​ (initial score: 8): Already positive, c3cP's concerns about overfitting and TripoSG selection were robustly addressed with zero-shot tests and human evaluations. Having expressed satisfaction in the rebuttal phase, they would likely maintain the score at 8, with possible reinforcement to a higher endorsement given the additional empirical support.

Reviewer EdXQ​ (initial score: 2): The rebuttal's generalizability demonstrations and case analyses partially countered the novelty and domain-restriction criticisms. However, EdXQ's fundamental issue with the work as engineering integration rather than research innovation might have persisted, leading to a modest increase to 4 but not full acceptance, as core conceptual gaps remain.

Reviewer UFR6​ (initial score: 8): Explicitly confirmed score maintenance after the rebuttal, citing satisfaction with responses. No change is expected; UFR6 would likely uphold the 8, considering the rebuttal adequately resolved questions about pipeline intuitiveness and benchmark challenges.

Reviewer 4ewk​ (initial score: 4): While the rebuttal offered human evaluations for TripoSG and zero-shot tests, concerns about metric coupling (RLVR rewards mirroring evaluation) and GPT-4o captioning bias were not fully resolved. 4ewk might have cautiously increased the score to  6 due to improved validation but would likely retain reservations about inherent biases and practical deployment costs.

---

### Decision · Program_Chairs · 2026-01-26

Accept (Poster)